# From ‘Farm to Fork’: Exploring the Potential of Nutrient-Rich and Stress-Resilient Emergent Crops for Sustainable and Healthy Food in the Mediterranean Region in the Face of Climate Change Challenges

**DOI:** 10.3390/plants13141914

**Published:** 2024-07-11

**Authors:** Javier Matías, María José Rodríguez, Antonio Carrillo-Vico, Joan Casals, Sara Fondevilla, Claudia Mónika Haros, Justo Pedroche, Nieves Aparicio, Nieves Fernández-García, Ingrid Aguiló-Aguayo, Cristina Soler-Rivas, Pedro A. Caballero, Asunción Morte, Daniel Rico, María Reguera

**Affiliations:** 1Agrarian Research Institute “La Orden-Valdesequera” of Extremadura (CICYTEX), 06187 Guadajira (Badajoz), Spain; javier.matias@juntaex.es; 2Technological Institute of Food and Agriculture of Extremadura (INTAEX-CICYTEX), Avda. Adolfo Suárez s/n, 06007 Badajoz, Spain; mariajose.rodriguezg@juntaex.es; 3Instituto de Biomedicina de Sevilla, IBiS/Hospital Universitario Virgen del Rocío/CSIC/Universidad de Sevilla, 41013 Seville, Spain; vico@us.es; 4Departamento de Bioquímica Médica y Biología Molecular e Inmunología, Facultad de Medicina, Universidad de Sevilla, 41009 Seville, Spain; 5Fundació Miquel Agustí/HorPTA, Department of Agri-Food Engineering and Biotechnology, Universitat Politècnica de Catalunya (UPC)-BarcelonaTech, 08860 Castelldefels, Spain; joan.casals-missio@upc.edu; 6Institute for Sustainable Agriculture, Consejo Superior de Investigaciones Científicas, Avda. Menéndez Pidal s/n, 14004 Córdoba, Spain; sfondevilla@ias.csic.es; 7Cereal Group, Institute of Agrochemistry and Food Technology (IATA-CSIC), Av. Agustín Escardino 7, Parque Científico, 46980 Valencia, Spain; cmharos@iata.csic.es; 8Group of Plant Proteins, Instituto de la Grasa, CSIC. Ctra. de Utrera Km. 1, 41013 Seville, Spain; j.pedroche@csic.es; 9Agro-Technological Institute of Castilla y León (ITACyL), Ctra. Burgos Km. 119, 47071 Valladolid, Spain; apagutni@itacyl.es; 10Department of Abiotic Stress and Plant Pathology, Centro de Edafología y Biología Aplicada del Segura (CSIC), Campus Universitario de Espinardo, 30100 Murcia, Spain; nieves@cebas.csic.es; 11Postharvest Programme, Institute of Agrifood Research and Technology (IRTA), Parc Agrobiotech Lleida, Parc de Gardeny, Edifici Fruitcentre, 25003 Lleida, Spain; ingrid.aguilo@irta.cat; 12Departamento de Producción y Caracterización de Nuevos Alimentos, Institute of Food Science Research-CIAL (UAM+CSIC), Campus de Cantoblanco, Universidad Autónoma de Madrid, C/Nicolas Cabrera 9, 28049 Madrid, Spain; cristina.soler@uam.es; 13Sección Departamental de Ciencias de la Alimentación, Facultad de Ciencias, Universidad Autónoma de Madrid, Campus de Cantoblanco, 28049 Madrid, Spain; 14Food Technology, Department of Agriculture and Forestry Engineering, Universidad de Valladolid, 34004 Palencia, Spain; pedroantonio.caballero@uva.es; 15Departamento Biología Vegetal, Facultad de Biología, Campus Universitario de Espinardo, Universidad de Murcia, 30100 Murcia, Spain; amorte@um.es; 16Department of Medicine, Dermatology and Toxicology, Universidad de Valladolid, Av. Ramón y Cajal, 7, 47005 Valladolid, Spain; daniel.rico@uva.es; 17Departamento de Biología, Campus de Cantoblanco, Universidad Autónoma de Madrid, C/Darwin 2, 28049 Madrid, Spain

**Keywords:** NUS crops, Mediterranean agrifood systems, abiotic stressors, agricultural diversification, nutritional quality, healthy diets, climate resilience

## Abstract

In the dynamic landscape of agriculture and food science, incorporating emergent crops appears as a pioneering solution for diversifying agriculture, unlocking possibilities for sustainable cultivation and nutritional bolstering food security, and creating economic prospects amid evolving environmental and market conditions with positive impacts on human health. This review explores the potential of utilizing emergent crops in Mediterranean environments under current climate scenarios, emphasizing the manifold benefits of agricultural and food system diversification and assessing the impact of environmental factors on their quality and consumer health. Through a deep exploration of the resilience, nutritional value, and health impacts of neglected and underutilized species (NUS) such as quinoa, amaranth, chia, moringa, buckwheat, millet, teff, hemp, or desert truffles, their capacity to thrive in the changing Mediterranean climate is highlighted, offering novel opportunities for agriculture and functional food development. By analysing how promoting agricultural diversification can enhance food system adaptability to evolving environmental conditions, fostering sustainability and resilience, we discuss recent findings that underscore the main benefits and limitations of these crops from agricultural, food science, and health perspectives, all crucial for responsible and sustainable adoption. Thus, by using a sustainable and holistic approach, this revision analyses how the integration of NUS crops into Mediterranean agrifood systems can enhance agriculture resilience and food quality addressing environmental, nutritional, biomedical, economic, and cultural dimensions, thereby mitigating the risks associated with monoculture practices and bolstering local economies and livelihoods under new climate scenarios.

## 1. Introduction

Considering the current trends in population growth, achieving global food security remains a future challenge in which agriculture plays a key role [1]. Currently, agriculture is based on the monoculture of a limited number of species [2]. This practice leads to increased fragility of the agricultural system (e.g., in the face of biotic and abiotic stresses), resulting in a depletion of available food ingredients, as our diet relies on a few plant species [3,4]. Furthermore, there is substantial scientific evidence indicating that more diverse agrifood systems are generally safer from both nutritional and agronomical perspectives, showing greater resilience to environmental changes and offering a broader array of nutritionally effective food products [3]. Indeed, growing a variety of crops enhances the nutritional diversity of the local diet. This is particularly important for certain communities that rely heavily on a few staple crops, helping address nutritional deficiencies and improve overall health [4,5]. Introducing novel crops will also offer farmers supplementary income streams, diminishing dependence on a singular crop and alleviating the financial risks associated with market fluctuations or adverse weather conditions [6]. Particularly, certain crops may be more resilient to drought, heat, or other climate-related challenges, providing a buffer against the impacts of climate change [7]. Also, diversifying crops into a rotation system helps break pest, weed infestation, and disease cycles, improves soil fertility, and reduces the risk of soil degradation [8] supporting sustainable agricultural practices. Likewise, the diversification of our agricultural system can contribute to minimizing food dependence that may be subjected, as we have experienced in recent times, to changes linked to international conflicts [9]. 

Currently, two-thirds of the plant-based foods we consume primarily come from three cereals, wheat, rice, and maize, providing 60% of dietary energy [10]. While the nutritional value of these crops is undeniable, they are deficient in essential nutrients crucial for ensuring a healthy and balanced diet, such as minerals, proteins, essential amino acids, or vitamins, among others [4,11]. Therefore, the diversification of our food crops (for human and animal consumption), by including crops with high nutritional value, is essential for achieving a balanced nutrient intake and promoting greater sustainability and moderation in the consumption of animal-origin foods [12]. 

The distinct climatic patterns of the Mediterranean region are characterized by mild, wet winters and warm, dry summers [13]. The unique climatic conditions and diverse ecosystems of these areas face increasing challenges derived from climate change which are linked to the escalating occurrence of droughts and elevated temperatures, which directly impact agriculture [14]. Also, pests and diseases may proliferate under altered climate conditions, posing additional challenges for farmers [15]. Furthermore, rainfed agriculture, which plays a pivotal role in southern and eastern Mediterranean countries, makes a substantial contribution to food security in the region [16]. Therefore, the Mediterranean agrifood system will require the development of adaptation strategies to cope with the aforementioned climatic changing conditions, giving special consideration to rainfed agriculture, particularly in light of its importance in relation to its extent in terms of surface area. This may involve the cultivation of more resilient crop species, which include neglected and underutilized species (NUS) and improved (bred) crop varieties, changes in farming practices, and the implementation of sustainable agricultural techniques to mitigate the impact of climate change, among others [10,17].

NUS, a novel and evolving category in agriculture, holds profound significance for global food security, particularly in addressing nutritional challenges. These crops represent a diverse array of plant species that exhibit resilience and adaptability. Their emergence is a response to the pressing need for sustainable and nutritious food sources, mostly when considering the challenges faced by conventional crops, such as susceptibility to pests, diseases, and climate change, especially in susceptible regions such as the Mediterranean [13,14,15]. They are promising alternatives for cultivation in diverse ecosystems and their adaptability can contribute to increased Mediterranean agricultural productivity [18].

Overall, it should be considered that the widespread adoption of agricultural diversification practices in the Mediterranean region can contribute to establishing a more sustainable and effective agriculture and food system at both local and global scales [19]. The cultivation of new crops can boost rural economies by creating employment, supporting local businesses, and contributing to overall economic growth. However, it is important to note that the successful introduction of new crops requires careful consideration of factors such as soil suitability, climate conditions, market demand, and farmers’ capacity to adapt to new cultivation practices, among others [17]. Additionally, supportive policies, extension services, and research initiatives play a vital role in facilitating the successful integration of new crops [20]. 

Taking all these aspects into consideration, this review aims to explore the potential of nutrient-rich and stress-resilient emergent crops as a viable solution to foster sustainable and healthy food production in the face of climatic challenges in the Mediterranean region. By examining the adaptability and nutritional value of selected NUSs (including quinoa (*Chenopodium quinoa* Willd.), amaranth (*Amaranthus* spp.), chia (*Salvia hispanica* L.), moringa (*Moringa oleifera* Lam.), buckwheat (*Fagopyrum esculentum* Moench and *Fagopyrum tataricum* (L.) Gaertn.), millet (including *Panicum miliaceum* L. or *Setaria italica* L., among others), teff (*Eragrostis tef* (Zucc.) Trotter), hemp (*Cannabis sativa* L.), or desert truffles (including *Terfezia claveryi* Chatin)), we aim to shed light on their role in mitigating the adverse effects of climate change on agricultural systems. This work not only delves into the agronomical features of these crops but also considers their nutritional, food technological, and health characteristics, emphasizing the importance of diversifying crops in the Mediterranean region for resilient and nutritionally rich food sources. Through this exploration, we aspire to contribute to the development of strategies that promote food security and sustainability in the Mediterranean region amidst the dynamic landscape of climate change. 

## 2. Agronomical Aspects Related to NUSs for the Mediterranean Region

### 2.1. Historical Background

Since the advent of agriculture, the Mediterranean region has been a centre of agricultural innovation and crop diversification [21]. Pedoclimatic conditions, characterized by a summer dry season and light soils, facilitated the introduction of Neolithic crop cultures centred around annual plants [22]. Early farmers inherited agricultural practices from the Near East, incorporating domesticated crops such as wheat and barley (*Hordeum vulgare* L.), along with legumes such as peas (*Pisum sativum* L.), lentils (*Lens culinaris* L.), and chickpeas (*Cicer arietinum* L.) [23]. This underscores the significance of these initial introductions over millennia. The prolonged history of co-evolution between these introduced plants, local agroclimatic conditions, and human preferences has shaped the Mediterranean area into an important secondary diversity centre [24]. 

Built on this foundation, throughout history, new plant species were introduced, being crucial for colonizing new croplands and enhancing food security. These can be considered the emerging crops of their respective eras. For instance, in early times, rye (*Secale cereale* L.), initially introduced as a weed, became a significant crop for expanding agricultural societies in northern Europe due to its competitive strength in poor soils and unfavourable climates [25]. In the Middle Ages, Arabic settlements introduced crops like rice, playing a vital role in developing agriculture in deltaic areas, and sorghum (*Sorghum bicolor* L. Moench), cultivated in marginal lands with insufficient rainfall, contributing to ensuring food production [26]. In more recent times, exchanges with the Americas introduced key species, notably maize and potato (*Solanum tuberosum* L.), swiftly integrated into agricultural systems, especially in regions facing challenges in producing bread-making cereals. 

Several other ‘emerging crops’ were successful in specific areas, highlighting the significance of plant introductions for local agriculture in the past. Many of these are now considered NUS but hold potential as alternative crops to enhance agricultural production in specific environments [27]. Some other examples are common millet (*Panicum miliaceum* L.), foxtail millet (*Setaria italica* L.), buckwheat, or different *Triticum* species such as einkorn (*T*. *monococcum* L.), polish wheat (*T*. *polonicum* L.), club wheat (*T*. *compactum* L.), or spelt (*T*. *spelta* L.) [28]. These plants played pivotal roles in food security during periods of agricultural crisis, caused by the depletion of soil fertility or climate crisis. A notable example of the role of emerging crops in adapting to climate changes is demonstrated by societies’ response to the Little Ice Age (LIA) cooling (1550–1710) [29]. This cooling period had adverse effects on agricultural production, leading to heightened famine. Societies that diversified agriculture, through the introduction of new crops and adapting cultivation techniques, exhibited increased resilience [30]. 

Apart from plant species, fungi have also been important for human nutrition. For example, fungal species have been harvested from the wild or cultivated to complement diets [31]. A notable example is the desert truffle (*Terfezia claveryi* Chatin), which has recently been integrated into Mediterranean cropping systems. Historically, desert truffles were gathered from the wild by diverse cultures across the Middle East, Africa, and Australia. Despite geographical differences, these cultures used similar methods for truffle collection, preparation, and use [32]. The cultivation of desert truffles began relatively recently, with the first plantation established in Murcia, Spain, in 1999. This effort led to the production of truffles two years later, marking the start of global desert truffle cultivation, which has since expanded to various regions in Spain and other countries [33,34].

In summary, from the origin of Mediterranean agricultural societies to the present, emerging crops have been fundamental to agricultural development and food security. Adapting agriculture to the current climate crisis should draw from past successful experiences, leveraging the agronomic potential of underutilized crops and the diversity of landraces that have co-evolved with Mediterranean agrosystems.

### 2.2. Importance and Benefits

Biodiversity holds paramount importance for ensuring food security and nutrition in Mediterranean agricultural ecosystems. The rich diversity within agricultural systems serves as a foundational asset for continually enhancing varieties essential for sustaining agriculture in the context of climate change. In line with this, the crucial role of NUS in sustainable agriculture, ensuring improved environmental protection and a consistent food supply, entails selecting cost-effective varieties adapted to current cultivation areas and of economic and agronomic interest [1]. Despite NUS encompassing crops that lack broad adoption, they offer added value compared to conventional crops, being nutrient rich, adaptable to poor soils and arid areas, requiring minimal water for proper development, and utilizing nearly 100% of the plant (root, leaf, seed, etc.) or truffle. 

In the face of climate change and evolving markets, knowledge and innovation become pivotal for sustainable success in agriculture [35]. Enhancing innovation performance becomes crucial for institutions and companies in the agricultural sector to remain competitive. Beyond technology adoption, innovation must anticipate market needs and identify superior products and processes, as exemplified by the introduction of new sustainable crops in Mediterranean areas such as Portugal, Spain, Italy, Greece, Turkey, Egypt, and Morocco [36]. 

Recognizing the significant negative impact of climate change on food production, the international scientific community acknowledges increased risks such as temperature fluctuations, droughts, and floods. The agricultural sector, a net emitter of greenhouse gases, faces vulnerability to climate change effects. Adapting, reducing vulnerabilities, and increasing resilience to climate change is thus crucial. The introduction of NUS crops emerges as a strategy to adapt and ensure food security and good nutrition in the face of climate change, aligning with the 17 Sustainable Development Goals committed by the global community [37]. However, climate change poses challenges to achieving these goals, hindering livelihood development and food security, especially for vulnerable populations. The incorporation of NUS becomes instrumental in overcoming these challenges and fostering food security and nutrition for humanity in the future [37].

### 2.3. Classification and Characteristics

NUSs are also referred to as novel or emergent crops, break crops, or alternative crops, among other terms in use. These crops can be classified based on various criteria such as use, climate adaptability, nutritional content, growth requirements, and other specific traits. According to their intended use, emerging crops can be destined for food, e.g., quinoa, or for industrial use, e.g., hemp. The classification can also focus on traits associated with climate adaptability, which make crops well suited for specific conditions (Table 1). Consequently, different categories can be established in this line, such as drought-tolerant crops (e.g., quinoa [38], millet [39], hemp [40], and desert truffles [41]), heat-resistant crops (e.g., millet [42], amaranth [43]), or salinity-tolerant crops (e.g., quinoa [44], sorghum [45]). Furthermore, crop growth requirements can be categorized into three levels: low, medium, and high. In the case of NUS, there is significant importance placed on cultivating crops with relatively low growth requirements. This emphasis is geared towards reducing the environmental impact of agriculture on climate change and enhancing the resilience of agricultural systems. Moreover, emerging crops can also be classified based on their nutritional value (See Section 3), establishing different categories such as high-quality protein crops (e.g., quinoa, amaranth, moringa, desert truffles), nutrient-dense crops (e.g., moringa, quinoa, amaranth), or omega-3-rich crops (e.g., chia, flaxseeds (*Linum usitatissimum* L.)), among others.

Quinoa, amaranth, millet, buckwheat, hemp, teff, moringa, desert truffles, and chia, among others (Figure 1), are considered underutilized climate-resilient crops [46,47,48,49,50,51,52,53] (Table 1). The introduction of these eco-friendly crops into our agriculture has the potential to enhance biodiversity and resilience in agricultural systems. Currently, agricultural production heavily relies on only a few species, and diversifying with these crops can contribute to a more sustainable and robust agricultural ecosystem [51]. These emerging crops showcase agronomic traits well suited to diverse environmental conditions, characteristic of each species. For instance, quinoa shows great potential due to its ability to withstand abiotic stresses growing in harsh environments with low fertility [54]. Its adaptability to diverse environmental conditions is attributed to its extensive genetic diversity [55], exhibiting tolerance to drought, frost, and salinity, while thriving on poor soils [54]. Amaranth, millet, hemp, chia, and desert truffle also demonstrated resilience to drought [39,40,56,57,58], making them low-water-requirement options for cultivation. Teff is a warm-season cereal crop adapted to a wide range of environments that can be grown under rainfed conditions [59]. Moringa exhibits tolerance to drought conditions. In regions with a subtropical climate, moringa can resist up to six months of dry seasons, provided the annual rainfall amounts to 500 mm [60].

Tolerance to heat has been reported for amaranth, a C4 plant, and millet [42,57], while millet, amaranth, hemp, and quinoa can withstand relatively saline soils [45,61,62,63]. Buckwheat thrives in cold climates and is well suited for poor soils, with reduced water requirements [64]. By contrast, certain desert truffle species from *Terfezia* sp. and *Tirmania* sp. face population decline due to reduced rainfall and changes in land use, prompting research on domestication for a drought-resistant crop in line with sustainable practices [33]. Overall, these minor crops can thrive in marginal soils.

The primary challenge hindering their expansion and introduction in regions like the Mediterranean basin is the scarcity of commercially available varieties well adapted to the specific environmental conditions of these areas. For example, photoperiod is the primary obstacle to quinoa’s adaptability in the northern hemisphere [65]. Quinoa is a facultative short-day and day-neutral plant [66]. Photoperiod is also a major limitation for chia cultivation in Europe because it requires a floral induction photoperiod of around 12 h of darkness [67]. Furthermore, quinoa and buckwheat growing under high temperatures during flowering can experience dramatic yield losses [50,68]. On the contrary, it has been shown that one of the primary constraints on the growth and productivity of moringa is low temperatures. Moringa thrives in tropical and subtropical climates. Moringa can withstand light frosts down to −3 °C. Additionally, it is a demanding crop in terms of light [60]. In the case of teff, lodging represents the most significant issue limiting yields [69], and drought adaptation in desert truffle mycorrhizal plants is key when aiming at expanding its cultivation and involves enhancing plant nutrition, photosynthesis, water use efficiency, and root colonization morphology [41]. Additionally, “turmiculture,” an agricultural approach with minimal water input and sustainable practices, enables the cultivation of this natural resource in arid and semi-arid areas [34].

Therefore, in the context of climate change, traits that make emerging crops well suited for Mediterranean conditions include tolerance to the harsh environmental conditions expected to intensify in the coming years such as drought, heat, and salinity [70].

**Table 1 plants-13-01914-t001:** A list of NUS crops with the potential of cultivation in Mediterranean environments.

Common Name	Scientific Name	Botanical Family	Type	Abiotic Stress Tolerance			
				Cold	Heat	Drought	Salinity
Quinoa	*Chenopodium quinoa* Willd.	Amaranthaceae	Dicotyledons/Herbaceous/C3	Medium	Medium/low	High	Medium/high
Amaranth	*Amaranthus* sp. (75 species [71])	Amaranthaceae	Dicotyledons/Herbaceous/C4	Medium/low	High	High	Medium
Buckwheat	*Fagopyrum* sp.	Polygonaceae	Dicotyledons/Herbaceous/C3	Medium	Medium/low	High	Medium
Millet/Sorghum	Subfamilies: - Chloridoideae - Panicoideae (11 species excluding teff) [72]	Poaceae	Monocotyledons/Herbaceous/C4	Low	High	High	High
Teff	*Eragrostis tef* (Zucc.) Trotter	Poaceae	Monocotyledons/Herbaceous/C4	Medium	High	Medium	Medium/low
Chia	*Salvia hispanica* L.	Lamiaceae	Dicotyledons/Herbaceous/C3	Low	Medium	Medium	Medium
Moringa	*Moringa oleifera* Lam.	Moringaceae	Dicotyledons/Woody/C3	Very low	High	High	Medium/high
Hemp	*Cannabis sativa* L.	Cannabaceae	Dicotyledons/Herbaceous/C3	Low	Medium/high	High	High
Desert truffles	*Terfezia claveryi* Chatin (with *Helianthemum almeriense* Pau)	Pezizaceae (Cistaceae)	Ascomycete (Dicotyledons/woody/C3)	Medium/high	Medium	Medium	Medium/low

### 2.4. Challenges and Limitations

The cultivation of NUSs poses various challenges across different facets of the agricultural and supply chain landscape. Ongoing threats to agricultural environments stem from a growing world population, limited arable land, and swift climate changes. Projections suggest that the global population will surpass 9 billion by 2050 [73]. The challenge of feeding this expanding population is a global concern. To secure food and ecosystem stability, it is essential to develop NUSs as crops for sustainable agriculture, emphasizing increased net production while minimizing adverse environmental impacts. Nevertheless, incorporating NUSs into existing rotations introduces complexities, necessitating careful consideration of interactions with other crops to prevent adverse impacts on soil health and pest dynamics. The utilization of diversified crop rotations is seen as pivotal in addressing sustainability challenges in modern agriculture. However, designing rotations that strike compromises across multiple sustainability domains remains a persistent challenge [74]. Pests and diseases pose another set of challenges, as resistant varieties for NUSs may be limited, rendering them susceptible to various threats. In this context, monitoring and managing pests and diseases become challenging due to the relative novelty of NUS in certain regions, where existing control measures may not be well established. Advancements in documenting pathogens and pests affecting NUSs, along with the development of corresponding management strategies, are, therefore, essential [75].

NUSs face an additional challenge due to the growing interest of consumers worldwide in aligning their food consumption with sustainability [76] and the increasing demand for healthier food choices [77]. These are the factors driving the increasing interest in nutrient-rich and stress-resilient NUS crops. Furthermore, processing emerging crops faces infrastructure and technological challenges. It is crucial to adapt existing facilities to the processing requirements of NUSs and concurrently establish quality control measures to meet market standards. Despite the great interest in NUSs, most are not yet well adapted and highly marketable. Developing efficient supply chains, encompassing logistics, distribution, transportation, storage, and distribution networks, is crucial [78]. Limited awareness among consumers and retailers necessitates marketing and educational initiatives. The specific niche markets associated with NUSs create challenges in establishing demand, requiring targeted marketing strategies. Socio-economic factors further complicate the adoption of NUSs. Convincing farmers to embrace change and cultivate NUS crops, especially when traditional crops have established markets and proven agronomic practices, is a hurdle. The economic feasibility of novel crops´ cultivation compared to traditional crops plays a pivotal role, requiring assurances of long-term financial viability. Additionally, the lack of supportive policies, such as research incentives or financial aid, can impede widespread adoption.

Thus, addressing all the above-mentioned challenges is essential for the successful integration of NUS crops into agricultural practices, ensuring sustainable cultivation, economic viability, and widespread adoption. Collaboration among farmers, researchers, policymakers, and industry stakeholders is crucial to developing sustainable cultivation practices, improving processing technologies, and creating effective market strategies. Additionally, investments in research and development, extension services, and capacity-building programs can contribute to the successful integration of emerging crops into agricultural systems and markets.

### 2.5. Genetic and Breeding Advances

Genetic improvement in NUS crops has been considerably lower compared to major crops. Consequently, there exists significant potential for enhancement in various agronomical and nutritional traits. Additionally, many of these crops originate from regions with agroclimatic conditions vastly different from those in the Mediterranean. Hence, ongoing breeding efforts are essential to develop varieties better adapted to Mediterranean agroclimatic conditions. As previously mentioned, photoperiod sensitivity poses a challenge for cultivating some NUS crops in the Mediterranean basin. Numerous accessions of quinoa, amaranth, chia, buckwheat, and hemp require short days for optimal production. For instance, quinoa exhibits two germplasm pools: the primary Andean highland quinoa and the secondary central and southern Chilean quinoa [79]. Fortunately, certain quinoa accessions showing day-length insensitivity have been identified in the coastal conditions of southern Chile. These have been utilized to develop varieties capable of thriving in European long-day environments. European varieties, developed in Denmark and the Netherlands, are currently available, and a breeding program is underway in Spain to create new varieties specifically adapted to Mediterranean field conditions [80]. Similarly, while many buckwheat varieties are sensitive to photoperiod, Europe now cultivates varieties that are not influenced by day length [81]. Regarding chia, there are a few patented varieties, such as the ‘Ouro’ variety from France, exhibiting long-day flowering characteristics, obtained through selective breeding [82]. Hemp, being a short-day plant with flowering highly dependent on photoperiod and temperature, also includes varieties like ‘Finola’ that can initiate flowering independently of photoperiod [83].

One of the key constraints for crop cultivation in the Mediterranean basin is the occurrence of episodes of high temperatures during the crop growing cycle. Numerous studies have demonstrated that elevated temperatures at flowering time can significantly diminish yield in quinoa and buckwheat [50,68]. Consequently, enhancing heat tolerance has become a crucial breeding target to increase the yield of quinoa and buckwheat in the Mediterranean basin.

The success of developing new crop varieties that thrive in different environments and exhibit enhanced agronomic and nutritional traits depends on genetic variability, understanding the genetics of target traits, and having effective improvement methods and tools. Quinoa germplasm exhibits high diversity, allowing its adaptation to various environments. However, accessing South American germplasm presents limitations, requiring negotiations with national governments and often necessitating facilitation by international organizations like FAO [84]. Despite being domesticated 7000 years ago by ancient Andean civilizations, quinoa breeding programs only commenced in the 1960s in the Andes and later in other regions from the 1970s onwards [85]. Several quinoa breeding programs, including hybridization, are ongoing in different countries. However, scientific knowledge about critical aspects of quinoa breeding, such as the inheritance of desirable agronomic and quality traits or the identification of molecular markers linked to them for marker-assisted selection, remains limited. The sequencing of the quinoa genome by Jarvis et al. (2017) [86] has provided a powerful tool to expedite quinoa breeding. For instance, this genome sequence facilitated the identification of a gene controlling saponin content [86]. Moreover, the availability of the quinoa genome sequence has enabled the identification, through genome-wide association studies (GWAS), of candidate genes for photoperiodic flowering regulation, thousand seed weight, and other agronomic traits [87,88]. Additionally, extreme gradient boosting has been employed to identify candidate genes potentially controlling betalain content in quinoa seeds [89]. These studies open the possibility of utilizing marker-assisted selection in quinoa breeding.

In contrast, a notable scarcity of genetic diversity has been observed within domesticated chia accessions, with slightly greater variability present in wild accessions [82]. Furthermore, gene banks hold limited chia germplasm [90]. Additional constraints for chia breeding include the presence of small and fragile corollas, fused flower parts, and a propensity for self-pollination, factors that have impeded the establishment of breeding programs centred on hybrids [91]. Chia has not been the focus of extensive modern plant breeding efforts, and improved cultivars have primarily arisen through the selection of lines from mixed germplasm sources, typically landraces [91]. Thus, limited knowledge exists regarding the genetics of pertinent traits in chia, although some progress has been achieved. Studies on the inheritance of photoperiod response, seed colour, lodging resistance, and shattering resistance have been conducted [91]. Also, transcriptomic studies have identified genes involved in secondary metabolites and oil biosynthesis [92]. Furthermore, chia genome sequencing has facilitated the characterization of genes associated with polyunsaturated fatty acids (PUFAs), mucilage biosynthesis, and those coding for oleosin, caleosin, and steroleosin [93].

Amaranth is considered an orphan crop due to the limited efforts directed towards its genetic improvement [94]. Nevertheless, some progress has been achieved through hybridization (including hybrids between different species), selection, and mutagenesis methods [79]. While most varieties have been developed using conventional breeding, gamma irradiation-induced mutation has also been employed in amaranth for the development of new cultivars [95]. Nonetheless, genetic studies in amaranth are scarce, with markers identified for plant growth, morphology, seed characteristics, and flowering behaviour. Inheritance studies have been conducted for nutritional factors, including starch characteristics, the perisperm layer of grain, seed protein content, and male sterility [79]. A Quantitative Trait Loci (QTL) for flower colour was identified in *A. hypocondriacus* L. [96]. Additionally, a recent GWAS in *Amaranthus tricolor* L. identified marker-trait associations associated with branching index, inflorescence colour, petiole pigmentation, and terminal inflorescence shape and attitude [97]. Noteworthy, genomic resources for amaranths include the genomes of several *Amaranth* spp. (the genomes of *A. hypocondriacus* L., *A. tricolor* L., *A. palmeri* S. Wats., and *A. cruentus* L. are available in the NCBI dataset), genome-wide SNPs and SSRs for *A. caudatus* L. and *A. hypochondriacus* L. and ‘popAmaranth,’ a population genetic genome browser for grain Amaranthus and their wild relatives [95].

Among all buckwheat species, only two are cultivated for grain food production: common buckwheat (*Fagopyrum esculentum* Moench) and tartary buckwheat (*F. tataricum* (L.) Gaertn). Common buckwheat has a wide distribution in Asia, Europe, America, and Austria, while tartary buckwheat is primarily grown in Asia [98]. Various breeding programs are dedicated to improving buckwheat varieties in Europe, where the cultivated types are short, early maturing, and not sensitive to photoperiod [81]. A crucial aspect of buckwheat breeding is that common buckwheat is highly allogamous due to the presence of unique dimorphic and sporophytic types of self-incompatibility, making it a heteromorphic strictly self-incompatible crop. The heteromorphic self-incompatibility in common buckwheat is controlled by a single gene locus [99]. A draft assembly of the common buckwheat genome, generated by Zhang et al. [100], successfully identified novel candidate genes controlling this trait. Additionally, a high-quality, chromosome-scale tartary buckwheat genome was released [101]. Progress has also been made in understanding the genetics of various traits in buckwheat. This includes studies on the genetic background of buckwheat flavonoid synthesis [102], QTLs for easy grain dehulling [103], and the identification of a gene involved in bitterness [104] which have contributed to advancing buckwheat breeding. Other authors have identified, by GWAS, candidate genes correlated with flavonoid accumulation and grain weight [100], as well as various morphological and yield-related traits [105] and rutin hydrolysis [106].

Millets, a group of small-grained, self-pollinated cereal grasses, include several species such as pearl millet (*Pennisetum glaucum* (L.) R.Br.), finger millet (*Eleusine coracana* Gaernt), proso millet (*Panicum miliaceum* L.), foxtail millet (*Setaria italica* (L.) P. Beauv.), little millet (*Panicum sumatrense* Roth ex Roem. & Schult.), and kodo millet (*Paspalum scrobiculatum* L.) or sorghum (*Sorghum bicolor* (L.) Moench) [107]. While millets naturally exhibit high genetic diversity, several hybridization methods have been developed to further enhance diversity [108]. The current discussion will focus on two millet species grown in Europe: foxtail millet and proso millet.

Foxtail millet, recognized for its high diversity, is emerging as a model organism for grasses due to its relatively small genome [107]. Molecular analyses suggest that China is the centre of foxtail millet diversity, with local landrace groups differentiating after domestication. China, Europe, and Afghanistan–Lebanon are proposed centres of domestication for this crop [107]. Molecular tools available for foxtail millet breeding include markers such as SNPs, ILP, and SSRs, used for genetic diversity studies and the mapping of various traits, including domestication and latitudinal adaptation traits, panicle morphology, inflorescence structure, branching and height, and agronomic traits [109]. Molecular facilities encompass databases for molecular markers, transcription factors, miRNA, and transposable element-based markers. Reference genomes and a pan-genome are also available [110,111]. GWASs have been conducted for agronomic and quality traits, as well as water absorption in seeds. In addition, transcriptomic studies have explored traits such as salt tolerance, drought, branching, freezing, and response to rust. Additionally, the expression of some NBLRR genes in response to *Magnaporthe grisea* (T.T. Hebert) M.E. Barr has been analysed [112].

Despite global breeding efforts and limited genetic characterization, proso millet remains an under-researched and underutilized crop, with only partial success achieved so far. Breeding methods for proso millet include pure line selection, pedigree selection, and backcrossing. The self-pollinating nature of proso millet has posed challenges in hybridization, resulting in only a few registered and released cultivars [107]. Genomic tools for proso millet are scarce, with RAPD, AFLP, SSR, and SNP markers, along with GBS and SPET sequencing, used for diversity studies revealing high diversity [113,114,115]. Also, a pan-genome is available [98], and a GWAS analysis for agronomic and seed traits has been conducted [116]. Gene expression studies include the identification of selenate and selenite transporters and their expression under salt stress and selenium, as well as a transcriptomic study investigating pathways associated with starch synthesis and metabolism to understand the effect and molecular mechanisms of nitrogen fertilization in starch synthesis in waxy and non-waxy proso millet varieties [117,118].

Teff has been cultivated in the Horn of Africa for millennia. Despite its high germplasm diversity, including wild relatives, landraces, and farmer varieties, this variability has not been efficiently exploited in breeding programs. However, Ethiopia has derived several varieties from farmer cultivars through mass selection or hybridization programs, resulting in increased yields. Chemical mutagenesis and CRISPR/Cas9 have been attempted to obtain varieties showing resistance to lodging, drought, increased seed size, and tolerance to herbicides, soil acidity, and salt [119,120]. Advances in marker-assisted selection have also been made with the sequencing of the teff genome [121]. QTLs have been identified for morphology and yield traits [122,123], and a GWAS approach used to study agronomic performance identified candidate genes for climatic adaptation and farmer appreciation [120]. Metabolomic studies for salt and drought tolerance, along with a transcriptomic analysis for calcium stress, have also been reported [124,125,126].

Hemp breeding has been intensive in Europe, resulting in a multitude of available varieties. Initially, breeders employed mass selection to enhance local landraces, followed by the introduction of foreign genetic material through crossing to augment fibre and seed yield [83]. In its natural state, *Cannabis sativa* L. is an annual dioecious plant, with a minor proportion of monoecious plants [127]. Dioecy is governed by two specific genes at linked loci, and the identification of female plants in early development is facilitated by the use of Y-specific DNA markers [128]. A pivotal moment in hemp breeding occurred with the discovery that fibre content is inherited from both the maternal and paternal sides in dioecious hemp. Another significant breakthrough was the identification of hermaphroditic monoecious plants capable of self-pollination, enabling the development of more uniform new varieties. Subsequently, the methods of self-pollination and crossbreeding of monoecious and dioecious plants became more widespread, contributing to the evolution of modern hemp varieties [83]. The identification of plants exhibiting photoperiod insensitivity marked another milestone. The molecular studies in hemp have primarily focused on the synthesis of secondary metabolites and disease resistance. Chemotype inheritance has been investigated [129], and genes involved in cannabinoid synthase have been mapped, with the identification of a gene responsible for cannabichromenic acid (CBCA) biosynthesis [130]. SNPs associated with genes in the cannabinoid pathway have been reported [131], along with QTLs linked to cannabinoid quantity and their ratios [129]. Genes related to resistance against powdery mildew have also been identified [132]. Also, a reference genome is available [133], which has facilitated the identification of genes involved in fatty acid and vitamin E biosynthesis.

Moringa is a cross-pollinated crop indigenous to India that exhibits both geitonogamy and xenogamy. The cultivation of this crop has expanded globally. Moringa varieties are classified into perennial and annual types. Perennial varieties are typically propagated from cuttings, while annuals are propagated from seeds. Both annual and perennial varieties are extensively utilized in breeding programs with a focus on various desirable traits, including dwarf stature suitable for leaf production, high-yielding types, elevated seed and oil content, and resistance to pests and diseases. Diversity genetic studies in Moringa have employed various molecular markers such as RAPD, SSR, CytP450, ISSR, SRAP, and SCoT [134]. Multiple reference genomes are available, with the most recent reported by [135]. These genomes have been instrumental in identifying genes responsible for high protein content, fast growth, and heat tolerance, along with heat shock transcription factors playing a significant role in defence against drought stress [136]. A transcriptome developed by [137] has further contributed to identifying genes responsible for the biosynthesis of secondary metabolites (such as quercetin, kaempferol, and benzylamine), vitamins, and iron transporters. Proteins involved in heat stress have also been reported [138].

## 3. Nutritional and Technological Aspects Related to NUSs for the Mediterranean Region

### 3.1. Nutritional Composition

Currently, diets largely depend on starch-rich staple crops such as maize, wheat, and rice [139]. Although these foods cover a large part of the daily caloric needs, they do not provide adequate amounts of other necessary nutrients and micronutrients [140]. Most agriculture-based solutions have not focused on improving nutritional quality but on food production [141]. This has resulted in neglecting the nutritional quality of crops and biodiversity, which negatively impacts human health. Therefore, there is a need to explore new or emerging crops with high nutritional quality (Table 2).

NUSs are often part of ancestral cultural traditions and have emerging value due to their unrecognized traits [142]. Recently, many minor crops have been marketed as health foods or “superfoods”, as was the case with quinoa (*Chenopodium quinoa*). The growing market interest in foods of high nutritional quality could represent the transition from a marginal crop to one of greater interest [143].

The growing interest in pseudocereals (such as quinoa, amaranth, and buckwheat) arises from their heightened protein content, improved amino acid composition, and balance when compared to traditional cereals, coupled with the presence of biologically active molecules [144]. Quinoa and amaranth species are esteemed for their richness in proteins [145], featuring a well-balanced essential amino acid profile [146], high levels of essential fatty acids such as linoleic (C18:2, ω-6) and α-linolenic (C18:3, ω-3) acids [147], and significant amounts of dietary fibre and minerals [148]. Similarly, buckwheat seeds stand out as a valuable source of protein and essential amino acids [149]. Chia, also recognized as an underutilized pseudocereal [150], boasts a nutritional profile encompassing essential nutrients such as proteins, omega-3 fatty acids, carbohydrates, dietary fibre, vitamins, and minerals [151]. Teff is nutritionally dense, featuring elevated levels of protein, carbohydrates, minerals, and dietary fibres [152]. Sorghum, on the other hand, provides slow-digesting starch for improved satiety and glycaemic response, along with a unique array of polyphenols with potential applications in reducing caloric impact and acting as a naturally stable food colouring source [153]. Conversely, leaves from minor crops such as moringa and hemp, distinguished by their rich nutritional constituents, including high protein, calcium, fibre, and vitamins, as well as low fat content, have garnered attention primarily for their non-nutritive phytochemicals [154]. Regarding desert truffles, although the protein levels are typically lower than their carbohydrate content, they are considered a healthy and hypocaloric product. This is not only due to their low fat content but also because most of their carbohydrates are classified as dietary fibres with structures different from those found in cereals and pseudocereals [155].
plants-13-01914-t002_Table 2Table 2Nutritional composition of selected NUSs with potential of cultivation in Mediterranean areas. Proximal compositional analysis has included protein, fat, ash, dietary fibre, and carbohydrate contents.Nutritional ComponentProtein g/100 gFatg/100 gAsh g/100 gDietary Fibreg/100 gCarbohydrates g/100 gReferencesQuinoa13.1–18.71.9–9.52.9–3.89.4–22.741.5–77.0[156,157,158]Amaranth13.0–17.64.2–8.52.4–410.0–25.048.0–69.0[156,159,160]Buckwheat8.5–18.81.5–6.51.7–2.720.0–26.054.5–57.4[161,162]Chía15–2525–404–4.818.0–35.526.0–42.0[156,163]Sorghum4.4–21.11.5–8.91.1–2.210.2–19.063.7–80.0[153,164]Teff9.1–12.62.3–3.32.4–2.52.4–9.870.8–73.8[165]Moringa18.6–39.41.4–5.27.9–8.22.0–10.229.8–51.6[166,167]Hemp21–3225.4–35.93.7–6.328.8–38.832.5–38.1[168]Desert truffle8–293–72.6–151.4–13.246.0–83.0[169]


In addition, all these crops are gluten-free and therefore suitable for celiac disease patients [170]. It should be considered that the nutritional composition of these crops depends on different factors such as climatic conditions, geographical location, soil attributes, and cultivation year. Therefore, the data provided in this review are ranges or average values found in the literature.

#### 3.1.1. Protein and Amino Acids

Plant proteins, particularly those derived from pseudocereals such as quinoa, amaranth, and buckwheat, have emerged as vital vegetable protein sources that not only impart health benefits but also contribute to environmental sustainability and enhance global food security [145,171]. The proteins in these pseudocereals exhibit high quality, characterized by a balanced essential amino acid composition [172]. Quinoa, with a protein content ranging from 13.1 to 18.7%, stands out for providing all nine essential amino acids (Table 2). Similarly, amaranth seeds boast a protein content ranging from 13.0 to 17.6%, closely rivalling wheat [173]. Buckwheat presents a protein content ranging from 8.5 to 18.8% (Table 2); however, the presence of antinutritional factors like tannins and proteases impacts protein digestibility [174]. Despite this, buckwheat proteins demonstrate a well-balanced composition of amino acids, with an abundance of lysine, typically limited in most cereals [175]. Chia seeds exhibit a protein content of 15 to 25% (Table 2), surpassing many traditional cereals in protein content [150]. These proteins are particularly rich in essential amino acids such as arginine, leucine, valine, lysine, and phenylalanine.

Cereals gaining recognition for their nutritional value include sorghum and teff. Sorghum samples display protein content ranging from 4.4 to 21.1% (Table 2). However, the presence of the hard-to-digest protein kafirin and a low lysine content reduces sorghum’s nutritional value for both humans and animals [176]. Teff seeds present a protein content range of 9.1 to 12.6% (Table 2) and exhibit a balanced essential and non-essential amino acid composition, with notable quantities of arginine, asparagine, and lysine, and lower amounts of leucine, phenylalanine, glutamic acid/glutamine, and proline [177].

In the case of *Moringa oleifera* Lam. leaves, consumed as highly nutritious vegetables, it stands out as a valuable source of plant protein, ranging from 18.6 to 39.4% (Table 2). These leaves contain all essential amino acids, positioning moringa leaf protein as a high-quality plant protein [166]. Similarly, *Terfezia* sp., *Tirmania* sp., or *Picoa* sp. truffles showed protein levels ranging from approx. 29 to 8% including sulphur-containing amino acids (cysteine and methionine) that generally remain limited in other plant-derived foods [178].

#### 3.1.2. Oils and Fatty Acids

Due to the rich quality and quantity of the lipid fraction in quinoa, amaranth, buckwheat, and chia, they are referred to as alternative oilseed crops. The oil content in quinoa varies from 1.9 to 9.5% (Table 2), with a rich source of essential fatty acids such as linoleic and α-linolenic acids. Oil content in quinoa seeds consists of approximately 88–90% unsaturated fatty acids (UFAs), out of which 66–68% are polyunsaturated fatty acids (PUFAs). Saturated fatty acids (SFAs) account for around 10–11% of the total fatty acids in the seed [171,179]. Linoleic acid is the main fatty acid present in quinoa seeds (58.5–61.4%) and has been proven to provide various health benefits such as preventing cardiovascular diseases [180]. The other major fatty acids in quinoa seeds are oleic acid (18:1) (17.5–20.5%) > palmitic acid (C16:0) (9.0–10.4%) > α-linolenic acid (5.1–7.2%) [171,181]. Furthermore, the ω-6:ω-3 ratio in lipid consumption is acknowledged as pivotal, as it is linked to various chronic diseases resulting from specific dietary patterns [147]. Low ω-6:ω-3 ratios are related to a low incidence of these diseases [182]. In quinoa, the ω6:ω3 ratio is found to be around 6–9:1 [171,179,183]. While this ratio is higher than the ideal one (4/1), it is much better than the typical ratio found in Western diets (15–16.7/1) [184]. On the other, amaranth seeds are also a source of high-quality fat (4.2–8.5%) (Table 2) [171]. Nevertheless, the composition of fatty acids varied compared to quinoa seeds, with lower linoleic acid content (35.2–37.3%). On the contrary, oleic acid (35.4–36.7%) was higher than that in quinoa. Important differences were also found between both species for palmitic acid (21% in amaranth seeds). The differences in the fatty acid profile extend also to the ω6:ω3 ratio, higher in amaranth (27.5–36.1/1) than in quinoa seeds [171].

Buckwheat’s lipid content is low (1.5–6.5%) (Table 2) but of great importance due to its composition. Buckwheat is a rich source of unsaturated fatty acids (74.5–79.3%) [185], primarily oleic and linoleic acids. These, along with palmitic acid, the predominant saturated fatty acid, constitute the most abundant components of the lipid fraction [186]. The α-linolenic acid is also present in smaller amounts.

Chia seeds are characterized by a high fat content (25–40%) (Table 2). They are particularly rich in polyunsaturated fatty acids (~80%), including α-linolenic acid (55.2–65.9%) and linoleic acid (17.8–22.1%), with a very low ω6:ω3 ratio (approximately 0.3–0.4) [163,187], making them a significant source of omega fatty acids. Palmitic acid is the principal saturated fatty acid, comprising 6.9–7.8% of the total lipids [163,188].

Sorghum seeds are claimed to have lower fat content when compared to other grains [189]. However, sorghum grains contain nearly 1.5 to 8.9% lipids (Table 2), with a composition based on linoleic acid (37.8–54.4%), oleic acid (25.8–43.5%), palmitic acid (14.1–20.8%), linolenic acid (0.98–2.3%), and stearic acid (C18:0) (0.99–2.2%) [190]. In contrast, polyunsaturated fatty acids are more susceptible to oxidation, reducing the shelf life and storage stability of fats and oils. In teff grains, the total fat content has been found within the range of 2.3–3.3 (Table 2), which was higher than the content in wheat (2.2–2.3%) and rice (2.9–2.9%) [191] but slightly lower than sorghum (3.6–3.8%), oat (3.7–6.4%) [191], and millet (3.4–6.5%) [192]. The predominant fatty acids are linoleic acid (17.3–39.9%), palmitic acid (17.6–43.4%), stearic acid (17.2–22.0%), oleic acid (9.98–17.4%), and α-linolenic acid (5.9–16.5%) [193].

In the case of moringa leaves and desert truffles carpophores, they show a low oil content (of 1.4–5.2% and 3–7%, respectively, Table 2); however, their lipid composition is outstanding. Moringa oils contain approximately 80% unsaturated and approximately 20% saturated fatty acids. Oleic acid is the most abundant (66–75%, approx.) [194], making moringa oil comparable to olive oil in terms of oleic acid. In truffles, oleic (32%) and linoleic (29%) acids are major fatty acids, as the latter is the precursor of the 1-octen-3-ol compound responsible for the characteristic ´mushroom´ aroma [195]. Among the different truffle species, *Picoa* sp. contains less palmitic and stearic acids but more oleic acid than *Terfezia* sp. [196]. Normally, vegetable oils from corn, rice bran, palm, peanut, safflower, and soybean contain less than 40% oleic acid. The high proportion of oleic acid in moringa oil renders it well suited for nutritional purposes and as a stable medium for cooking and frying [197]. Additionally, the presence of palmitic (~6%), stearic (~5–6%), and behenic (~4.4%) fatty acids has been confirmed in previous studies [198,199].

#### 3.1.3. Ash and Mineral Content

The ash content in a food item represents the total amount of minerals present, holding significant importance for various biological activities that benefit the human body. However, in vegetable and truffle crops, the concentrations can vary widely depending on the type of soil where the plant or fungus is grown and the fertilization rates applied [148].

Among the pseudocereals, quinoa, and amaranth seeds are good sources of minerals [148]. In general, the ash content of quinoa seeds (2.9–3.8%) (Table 2) is greater than that of rice, wheat, and other grains [157,200]. In the same way, the ash content in amaranth seeds ranged from 2.4 to 4% (Table 2). The main minerals found in these pseudocereal seeds are potassium (K), phosphorous (P), calcium (Ca), and magnesium (Mg), with amaranth being the plant species that typically shows the highest amount of minerals followed by quinoa [200]. In general, higher contents of P, Ca, and Mg and lower K contents are found in amaranth compared to quinoa [171]. K, the major mineral in both types of seeds, is an important chemical element that helps to prevent muscle weakness, respiratory insufficiency, and hypotension in humans [201]. On the other hand, Fe and Ca are crucial minerals in the diet [202]. Ca contents were up to three times higher in amaranth than in quinoa [203] and higher compared to other vegetable food products considered rich in Ca, such as almonds or legumes [204]. However, quinoa seeds contain more Fe than traditional grains, although saponins and phytic acids present in the seeds could affect their bioavailability to some extent [157].

Buckwheat grains showed a total mineral proportion between 1.7 and 2.7% (Table 2), slightly lower than the other pseudocereals, such as quinoa and amaranth. In general, P, K, Mg, and Ca are found at reasonable levels, while Fe, manganese (Mn), and Zinc (Zn) have lower concentrations in buckwheat [205]. Other authors found high levels of Mg, Ca, Fe, Zn, Mn, and Cu, while the contents of selenium (Se), molybdenum (Mo), and cadmium (Cd) are comparatively low [206]. Additionally, it has been reported that buckwheat could be used as a dietary nutrient source for Mg, Zn, Cu, Mn, and Se [206,207]. On the other hand, chia seeds as well as desert truffles are also a good source of minerals, nearly 4.0–4.8%, out of which K, Mg, Ca, and P are present in the highest amounts. In chia seeds, Fe is also present in substantial quantities, whereas in truffles its concentration varies significantly [163,178,188,195]. Ca is the most abundant microelement found in chia seeds, followed by P, Mg, and Na, according to the results obtained by [163]. The concentration of Ca in chia seeds is higher than that found in milk, and Fe concentration is higher than that found in lentils, spinach, and liver [208,209].

Sorghum and teff grains contain a relatively low ash content (1.2 and 2.5%, respectively) (Table 2). Sorghum contains a substantial amount of various minerals such as K, Mg, P, and Ca [210,211], with a bioavailability of 90% for Na and K. Thus, it provides a significant concentration of these elements for the human diet. Regarding teff grains, they present higher contents of Fe, Ca, and Cu than other cereals which is probably due to the small plant size and increased contact with soil [212]. Other authors also found higher concentrations of K followed by Mg, Ca, Na, and Fe [213]. The elevated levels of these minerals in teff grains may result from the ready availability of these elements in the soil, facilitating their uptake by the plant and subsequent translocation to various plant parts [214]. In addition, teff is noted for its high Fe bioavailability [215], which is associated with a lower prevalence of Fe-deficiency anaemia in populations where teff is consumed regularly.

Moringa leaves showed ash contents of around 8% (Table 2), including K, Na, Mg, and Fe as the predominant minerals, in decreasing amounts. The high value of ash indicated that moringa leaves are a good source of mineral elements. Thus, the contents of these minerals in moringa leaves could satisfy the recommended daily intake (RDI) [216]. Furthermore, it was reported that the mineral content in moringa leaves is significantly positively correlated with the elemental mineral concentration of the soil, and differences in the stage of growth of the moringa leaves could also contribute to the variations of the elemental mineral concentrations [217].

#### 3.1.4. Fibre and Carbohydrates

The pseudocereals in whole grain form are considered a rich source of dietary fibre [218]. The total dietary fibre (TDF) content in different varieties of quinoa, amaranth, and buckwheat ranged from 9.4 to 22.7%, 10 to 25%, and 20 to 26%, respectively (Table 2). In the case of buckwheat grains, they show more fibre than other pseudocereals [219], and these values surpass those obtained for other grains such as wheat, rice, and corn [173].

Chia seeds contain 18 to 35.5% dietary fibre (Table 2), placing them at the top of the list in providing more fibre. The TDF content of sorghum and teff grains differs, with contents between 10.2 and 19% and 2.4 and 3.9%, respectively (Table 2), found in the literature. However, higher values (8.6 and 9.8%) [220] were also reported for teff grains, likely associated with different methodologies or varieties used. Additionally, the fibre content in moringa leaves and desert truffles was relatively lower compared to most forage plants being 2–10.2% for moringa, 1.4–12.6% for *Terfezia* and 6.5–13.2% for *Tirmania* species. The dietary fibres in truffles differ from those in other plant crops, primarily comprising chitins and β-glucans with distinct branching boundaries compared to cereals (Table 2) [58,155,221].

On the other hand, carbohydrates are essential for the proper functioning of the body and should make up 50% of our daily calorie intake [221]. In quinoa, amaranth, buckwheat, and desert truffles, carbohydrates were the major macronutrients. The literature consulted found values between 41.5 and 77.0%, 48 and 69%, 54.5 and 57.4%, and 46 and 83%, respectively (Table 2), equal to or slightly lower than other cereals such as wheat, rice, and corn grains (71.7, 81.7, and 74.3 g 100 g^−1^, respectively) [173,178,222]. However, chia seeds showed a lower carbohydrate content (26–42%) (Table 2) compared to the other pseudocereals. It should be noted that a lower average carbohydrate content in chia seeds could be due to the high values obtained for all other nutritional parameters. Thus, since carbohydrate contents of plant food are calculated by difference, an increase in protein, fat, and moisture contents will ultimately affect the carbohydrate contents and contribute to observed increases. Sorghum and teff grains and moringa leaves are also rich sources of carbohydrates, with contents found in the literature ranging from 63.7 to 80%, 70.8 to 73.8%, and 29.8 to 51.6%, respectively (Table 2). Importantly, the main carbohydrate component of all crops studied is starch, constituting around 50–70% of it [157,223].

### 3.2. Bioactive Compounds

The widespread utilization of pseudocereals is attributed to their favourable nutritional value and the presence of bioactive compounds. These compounds, notably flavonoids, phenolic compounds, and peptides, exhibit numerous beneficial effects on human health [224]. However, the extraction conditions significantly impact the quantity of bioactive compounds extracted (solvent, extraction time, and temperature), leading to substantial variations in results reported in the literature. Consequently, this literature review refrains from providing specific values for each of the compounds found in different crops.

#### 3.2.1. Phenolics

Phenolic compounds, characterized by hydroxyl groups on aromatic hydrocarbon rings, constitute secondary plant metabolites. This compound family includes phenolic acids, flavonoids, phenols, quinines, coumarins, lignans, phenylpropanoids, and xanthones [225], existing in both free and bound forms. These compounds, particularly flavonoids, phenolic compounds, and peptides, confer various health benefits through antioxidant capacity, scavenging free radicals, and preventing oxidative stress [150,172].

Pseudocereals, like quinoa, amaranth, and buckwheat, exhibit rich concentrations of bioactive phytochemicals. Quinoa, in particular, boasts flavonoids such as quercetin and kaempferol, contributing to antioxidant properties and health benefits [200,226]. Buckwheat stands out as the richest source of phenolic acids, predominantly found in free forms, influencing antioxidant activity [227]. Chia seeds contain significant phenolic compounds, including various flavonoids, but also diterpenoid phenolics such as rosmarinic acid, both contributing to the antioxidant properties of these seeds [224,228]. Sorghum, uniquely rich in bioactive constituents, presents high concentrations of phenolic compounds such as ferulic and protocatechuic acids, as well as flavonoids like quercetin and kaempferol [229,230]. Teff grains prominently feature ferulic acid and flavone derivatives, with additional phenolic compounds such as protocatechuic and cinnamic acids [231]. Simple phenolic compounds and flavonoids have been extensively documented in desert truffles, as well as in edible mushrooms. However, there are conflicting reports regarding the ability of these fungi to either absorb or synthesize these compounds, especially flavonoids [195,232].

In summary, high-value seed flours, including those from buckwheat, quinoa, sorghum, and teff, exhibit elevated levels of phenolic substances and flavonoids, surpassing whole-grain cereal flours in antioxidant and anti-inflammatory effects [233,234]. Moringa leaves are rich in phytochemicals, with phenolic acids, flavonoids, and anthocyanins contributing to their nutritional profile [235,236].

#### 3.2.2. Phytosterols

Phytosterols, lipophilic compounds structurally akin to cholesterol, play a vital role in cholesterol metabolism and exhibit potential health benefits against cancer, diabetes, hepatoprotective effects, and cardiovascular diseases through their cholesterol-lowering characteristics [150]. In quinoa seeds, major phytosterols include β-sitosterol, campesterol, and stigmasterol, with higher quantities than those found in barley, rye, millet, and corn [180,237]. Amaranth seeds also contain these phytosterols, with β-sitosterol as the predominant compound [238]. Interestingly, according to [239], β-sitosterol, a significant phytosterol, may possess anti-diabetic properties.

Buckwheat grains feature β-sitosterol as the primary phytosterol, along with others like campesterol, avenasterol, and D-7-stigmasterol. Additionally, the infrequently observed cycloartenol was identified in buckwheat samples [240]. Chia seeds similarly contain phytosterols such as campesterol, stigmastanol, stigmasterol, D5-avenasterol, and β-sitosterol [241]. Sorghum grains encompass common sterols, including β-sitosterol, campesterol, and stigmasterol, positioning sorghum as a rich source of phytosterols compared to fruits, vegetables, and other cereals [230].

While there is limited evidence in the literature for the presence of phytosterols in teff grain, one study reported the existence of β-sitosterol and β-sitosterol-3-O-β-D-glucoside in teff grains [242]. Also, moringa leaves have emerged as a rich source of phytosterols, with β-sitosterol being the major compound, followed by stigmasterol, campesterol, and Δ5-avenasterol [154].

Fungal sterols in desert truffles remain inadequately investigated. Some reports suggest that the *Terfezia* genus predominantly contains brassicasterol (98%), with ergosterol constituting less than 2%. However, ergosterol is a major constituent in fungal membranes [243].

#### 3.2.3. Betalains

Betalains, nitrogen-compound plant pigments, have garnered attention in functional foods due to their bioactive properties, including antioxidant and anti-inflammatory effects [244,245]. Betalains demonstrate superior antioxidant activity compared to polyphenols and are exclusive to plants of the order Caryophyllales, including certain cultivars of the family Amaranthaceae [246].

Quinoa seeds derive their yellow, black, and red colours from betalains, making quinoa a potential crop for betalain extraction, although some varieties may lack measurable amounts of betalains [244]. Betalains are further categorized into red-violet betacyanins and yellow-orange betaxanthins [247]. The incorporation of these pigments in food products is officially certified by the European Union (additive E-162) and the US FDA [248]. Amaranthine and isoamaranthine are two betacyanins found in amaranth, while red and black quinoa seeds contain betacyanins such as betanin and isobetanin [249]. Some studies report low amounts of betalains in specific amaranth varieties [250].

### 3.3. Antinutritional Factors

Certain phytochemical compounds found in seeds, such as phytates, saponins, lectins, and protease inhibitors, can exert both positive and negative effects on human health. At low concentrations, these compounds may contribute to well-being by exhibiting antioxidant properties, anti-inflammatory effects, and anti-cancer or immune-enhancing effects. However, when present in higher concentrations, these phytochemicals can interfere with the absorption of essential minerals, induce haemolysis, disrupt protein digestion and absorption, or harm the gut wall, potentially leading to increased intestinal permeability. While a moderate intake of these compounds offers health benefits, excessive consumption should be avoided to maintain a balanced diet and ensure optimal nutrient absorption. Additionally, oxalates and nitrates, naturally occurring compounds in various plant foods, are generally considered harmless when consumed in moderation. Overall, various antinutritional factors are present in the edible parts of NUS crops, and their influence should be considered during the assessment of nutritional characteristics. In this regard, it is crucial to highlight the very limited knowledge about the presence of antinutrients in the majority of the NUS crops analysed in this work. This underscores the necessity of conducting additional research on this topic. Nonetheless, a concise analysis is presented herein, and the findings are summarized in Table 3. Furthermore, to mitigate the presence of antinutrient compounds, various actions or strategies can be employed during food processing, as outlined in Table 3.

In the case of phytic acid, also known as myo-inositol hexakisphosphate, it is commonly found in grains and seeds as phytates. While phytic acid intake has reported benefits such as antioxidant function and the prevention of heart diseases and cancer, it is notorious for inhibiting the absorption of minerals, proteins, and trace elements [251,252]. The formation of insoluble complexes in the gastrointestinal tract renders them non-absorbable, while enzymatic degradation of phytic acid by phytases improves the nutritional value of cereal, pseudocereal, legume, and oilseed products [253,254,255].

Saponins, characterized by a linear arrangement of hexose or pentose glycoside units linked to sapogenin aglycones, form complexes with proteins, lipids, zinc, and iron, exhibiting a haemolytic effect, albeit absorbed in limited quantities; they can enhance membrane permeability, potentially aiding in intestinal food intake and drug assimilation [256,257]. However, their bitter taste poses a challenge to the widespread utilization of quinoa or amaranth seeds [258].

Another antinutrient compound family, protease inhibitors, abundant in legumes, cereals, pseudocereals, and oilseeds, forms stable complexes with proteolytic enzymes, influencing human physiology in both beneficial and detrimental ways [259,260]. While they are considered antinutritional factors due to their inhibitory effects on digestive enzymes, quinoa and amaranth seeds contain minimal amounts, unlike commonly consumed grains [256]. Conversely, the limited digestibility of buckwheat protein is attributed to various factors, including protease inhibitors, tannins, and α-amylase inhibitors [162]. Oxalates, found in various plant tissues including leaves, stems, hypocotyl–root, and seeds, hinder the absorption of calcium and magnesium, contributing to kidney stone formation. Amaranth and quinoa are considered high oxalate sources, with the highest concentrations observed in leaves and stems. However, the majority of oxalates are found in an insoluble form, and due to their high concentrations of calcium and magnesium, this may result in low oxalate absorbability [261]. Nonetheless, to mitigate the risk of stone formation, limiting oxalate consumption to below 100 mg per day is advisable, with individuals prone to kidney stones potentially requiring further reduction to less than 50 mg per day [262]. Cooking grains before consumption is essential to reduce soluble oxalate intake, as soluble sources are more readily absorbed, posing a risk for kidney stone development [263]. In the case of lectins, whose toxicity is contingent upon the quantity consumed and the frequency of ingestion, they are proteins that can bind to cells, leading to structural and physiological damage in tissues upon consumption, resulting in potentially adverse health effects [264]. They have been found in quinoa, amaranth, or desert truffle [265,266,267].

Nitrates, while not inherently harmful, can convert to nitrites during digestion, potentially leading to the formation of nitrosamines in the stomach [268]. The World Health Organization (WHO) has not specified a recommended daily allowance (RDA) for nitrates, but the European Food Safety Authority (EFSA) has set an acceptable daily intake (ADI) at 3.7 mg/kg of body weight/day, approximately 260 mg/day for a 70 kg adult. Generally, grains or seeds from selected NUS crops fall below this threshold. However, given the lack of information on nitrate content in the green parts of these crops, it is advisable to conduct nitrate content analyses when consumed as vegetables, as vegetables are a primary source of nitrates in plant-based foods.

**Table 3 plants-13-01914-t003:** Mainly antinutritional factors in selected crops and the treatments to inhibit their action.

Antinutrient Factor	Phytic Acid	Saponins	Protease Inhibitors	Lectins	Oxalates	Nitrates	References
Crop and Usual Treatment	g/100 g	mg/100 g	mg/100 g	g/100 g	mg/100 g	mg/100 g	
Quinoa	0.50–1.39	1.90–6.24; 199.07 629.7–1633.3 **	2.4 * ^g/kg^	5.2 ^HU’^; 9.3%	4.31; 6.17–9.45	21.83	[158,258,266,269]
Amaranth	0.61–1.34	5.33–10.66 ^HU/mg^ 79–186	0.519–1.016 ^TIU/mg^	0.5–7.3; 11.2–130.6 ^HAU/mg^	178–278; 46–152	48–84	[258,261,267,270,271]
Buckwheat	0.63; 1.7	0.058	55 ^α U/mg protein^; 94 ^α U/mg protein^	NYR (+)	93–155	NYR (−)	[161,272,273,274]
Millet	0.12–0.92; 0.336–0.649; 0.519–0.784; 0.64–0.81	0.039–0.044	0.385–0.985 ^TIU/mg flour^	NYR (−)	11.3–20.0	NYR (−)	[164,272,275,276,277]
Teff	0.682–1.374; 1.35	722	NYR	NYR (−)	3.50–5.27 ***	NYR (−)	[278,279,280,281]
Chia	1.5–2.6	616	445 ^TIU/mg protein^	NYR	NYR (−)	NYR (−)	[163,282]
Moringa (seeds)	0.115 ^δ^;2.54	1080 ^δ^	15.4; 120 ^δ^	NYR (+)	2.21 ^δ^	NYR	[283,284]
Hemp	2.66–3.08	123.54 ^α^	3.6% ^β^	NYR	NYR	NYR	[285,286,287]
Dessert truffles	0.581	12.38	NYR	18.11 ^HU^	NYR	NYR	[265,288]
Usual Treatment to diminish the antinutrient factor concentration or deactivation. In general, antinutrient levels can be controlled by cultivar selection and breeding, field management, and processing	Soaking, Steeping, Malting, Fermentation, Germination, Extrusion, Popped grains, Phytase activation by sprouting, Germination, or lowering the pH Inclusion of exogenous phytases Agronomic conditions Genetic engineering	Germination Popped seeds Extrusion Sweet varieties (<110 mg/100 g) Agronomic conditions	Cooking around 100 °C or at higher temperatures Popped grains Extrusion	Cooking around 100 °C or more Soaking Extrusion Popped seeds Germination	Water cooking-soluble oxalate, Soaking/Steeping Control of fertilization/soil	Water cooking Soaking/Steeping Control of fertilization/soil	

* g/kg: the amount of enzyme inhibited per kg seed meal; HU’: Haemagglutinating Units × 10^−6^ per kg of seed meal, treated rat blood cells; TIU/mg, HAU/mg, and HU/mg: Trypsin Inhibitory Units, Haemolytic Activity Units, and Haemolytic Units per mg of the sample on a dry weight basis. ** expressed in sapogenins (the hydrolysed product of saponins); *** the amount in straw; ^δ^ defatted seed flour; ^α^ in the extract; ^β^ trypsin inhibitor; NYR not yet reported; presence: (+); apparently at a lower amount than other grains/seeds: (−).

### 3.4. Nutritional Challenges and Considerations

Currently, the global community confronts a spectrum of nutritional challenges encompassing malnutrition and hunger, juxtaposed with concerns regarding obesity and diet-related ailments such as diabetes and cardiovascular diseases. Concurrently, the exigencies of climate change and water scarcity necessitate attention, alongside underutilized food wastes, byproducts, and the inadequacy in meeting the protein demand of the world population [289]. In this regard, it becomes necessary, and almost mandatory, to leverage and capitalize on the scientific and cultural resources and knowledge at our disposal. Hence, aligning with the central theme of this review, the possibilities presented by the utilization and revaluation of NUSs are well recognized. These possibilities contribute to enhanced biodiversity and sustainable development, not only at the agronomic level but also within the industrial context. The ultimate goal is to achieve improved production, enhanced nutrition, a healthier environment, and an overall better quality of life for all, leaving no one behind [290].

Beyond popular labels such as ‘superfoods’ or ‘miracle seeds or trees’, the crops discussed in this work exhibit a chemical composition in macro- and micronutrients that are either similar to or superior to the four main plant crops dominating the global agrifood system—namely, rice, wheat, soybeans, and corn. Notably, the production of these major crops is concentrated in a few countries globally (USA, China, India, Argentina, Brazil, and Russia). A distinctive advantage of many emerging crops highlighted in this study is their superior adaptability to abiotic stress conditions, as detailed in Section 2 on agronomical aspects related to NUSs for the Mediterranean region [10,291]. Hence, it is imperative to undertake more comprehensive studies on the introduction and advancement of these emerging crops, involving genetic improvement, hybridization, artificial intelligence, and enhanced control over abiotic stresses and pests. The objective is to diversify both the plant raw materials dedicated to food and the national and local production levels. This strategic approach aims to mitigate recent food crises, such as those experienced during the COVID-19 pandemic or conflicts in regions like Ukraine and Gaza. The acquisition of greater agronomic knowledge will empower governments to implement agricultural programs and policies that positively impact nutrition.

Conducting research and disseminating findings on the nutritional aspects of emerging crops, including food composition, dietary assessments, and human nutritional requirements, is essential for integrating these crops into society and highlighting their health benefits. This effort addresses the crucial nutritional challenge of meeting the rising global protein demand through sustainable production, diversified protein sources, and efficient resource use.

As stated in Table 2, the seed protein content shown by some can exceed 20% (e.g., chia [150] or hemp [292]). Furthermore, all exhibit higher protein contents than commonly consumed cereals [145,171,176]. In this context, an underutilized and valuable resource to be revalued and exploited is the protein contribution from other parts of the plant besides the seeds, notably the leaves. Crops like moringa, for instance, present protein values ranging between 16 and 40% in this regard [166] or desert truffles, with values ranging between 14 and 29% [178,195]. The protein content can be further enhanced through the extraction and commercialization of other components applicable in the food industry, such as oil, starch, or fibre fraction. This process results in protein concentrates or isolates [293], which can serve as plant-based ingredients in products with high protein content catering to vulnerable populations such as the elderly, malnourished children, or pregnant women.

Recent advances in food and health emphasize the potential of peptides, mono- and polyunsaturated fatty acids, pigments, and low glycaemic index starches from these crops, suggesting promising research directions. This progress requires enhancing and updating extraction technologies, such as microwaves, ultrasound, high pressures, fermentation, tribo-electrostatics, and supercritical fluids. Additionally, there is ongoing exploration and utilization of extraction methodologies, such as those assisted by enzymes or eutectic solvents [294,295,296,297,298,299].

Another significant nutritional challenge associated with these crops is the utilization or revaluation of antinutritional compounds, particularly found in the seeds. Examples, as described earlier in this review, include saponins in quinoa [300], tannins in sorghum [301], or phytic acid in amaranth, chia, and millet [302]. Initially utilized by the plant as a defence or survival mechanism against pests, predators, or adverse weather conditions, these components were traditionally deemed exclusively antinutritional for decades. However, recent scientific studies have shifted the focus, attributing certain health benefits at low concentrations, such as antioxidant or anti-inflammatory properties [303,304,305]. The development of a more friendly and profitable technology with these raw materials at the industrial level will increase this revaluation and help to enhance the benefits of these crops at the food level.

### 3.5. Technological Aspects

The surge in quinoa demand is attributed to its distinctive nutritional and techno-functional characteristics. Formerly stigmatized as a symbol of rural poverty and indigeneity, quinoa has gained superfood status [306]. Notably, quinoa starch, with its high peak viscosity, holds promise for enhancing techno-functionality in gluten-free dough formulations and reinforcing edible biodegradable films. Research by Cordoba-Cerón et al. [307] showcased improved techno-functional properties in protein-rich pasta when incorporating defatted high-protein quinoa flour. Fungal fermentation has emerged as a viable method to enhance the bioactivity of flours, as demonstrated in quinoa, leading to increased levels of resistant starch and fibre [308]. Amaranth protein surpasses cereals in biological value and exhibits favourable techno-functional properties, including high solubility and foaming capacity. Studies by Kierulf et al. [309] demonstrated that amaranth and quinoa flours, retaining a portion of their protein content, possess excellent techno-functional properties, qualifying them as natural alternative surfactants for stabilizing food emulsions [309].

Chia seeds, endowed with intrinsic technological characteristics such as high water-holding capacity and robust gelling properties, are suitable as fat replacers in baked and processed meat products [310,311]. While other NUSs hold nutritional interest in food formulation, their technological applications present challenges. Hemp seed protein (of approximately 20–25%), characterized by low allergenicity and a complete amino acid profile, faces limitations in techno-functional properties for successful formulation in food products [312]. Extrusion technology has proven successful in overcoming these limitations, particularly for products mimicking meat [313]. Moringa encounters obstacles in food formulation due to organoleptic characteristics such as an astringent bitter taste and unpleasant smell. Encapsulation technology has been proposed to address these challenges while enhancing thermal stability [314]. Alternatively, the use of moringa flower buds in powdered format has been suggested for direct incorporation into pasta products, enhancing nutritional profiles and antioxidant activity [167].

Desert truffle (*Terfezia arenaria*), although not listed as an edible food in the European Union, presents an intriguing opportunity for sustainable plant-based meat analogues due to its unique aroma and composition analogous to meat [315]. Nevertheless, desert truffles have been successfully employed to enhance the nutritional composition and antioxidant content of biscuits [169].

#### 3.5.1. Culinary and Processing Aspects

The comprehensive examination of the culinary and processing characteristics of nutrient-rich and stress-resilient emergent crops provides insights into their potential applications in traditional cuisines and their nutritional benefits for promoting their consumption in conventional culinary practices and improving nutraceuticals for high-quality functional foods [316]. Quinoa, for example, is extensively utilized as a grain or flour in the preparation of bread, soups, and cooked products, either independently or in conjunction with other flours [54]. Given its compatibility with traditional cuisine culture, quinoa seamlessly integrates into South American dishes, lending its nutty flavour and fluffy texture to traditional Peruvian quinoa stews and contemporary salads, showcasing adaptability across diverse global cuisines [317]. Moreover, expanded or puffed quinoa introduces another variation of traditional quinoa products. This involves subjecting the grain to a high temperature and pressure, resulting in a quinoa pop. High temperatures and long processing times, common in energy bars and instant cereal production, can cause a decrease in protein and linoleic acids, impacting the nutritional quality of the product [318].

Chia seeds stand out for their flavour and impressive water-absorbing properties, offering a versatile, gel-like consistency that makes them a favoured ingredient in beverages, puddings, smoothies, and various sweet and savoury recipes for a nutrient-rich boost [319]. Their combined nutritional and technological attributes are optimal for use in baked goods as an egg replacement ingredient, or in the production of gluten-free bread or pasta, owing to their hydrocolloid and thickening properties [320]. Numerous studies have explored the incorporation of chia seeds and chia mucilage gel as substitutes for eggs, oil, or fat in diverse food products resulting in more nutritious items with acceptable sensory qualities, especially evident in pound cakes, reduced-fat bread baking, and improved quality in sweet wheat doughs [321,322,323]. Additionally, plant-based meat analogues offer an innovative strategy for emulating the sensory and nutritional attributes of traditional meat products. With an increasing demand for these alternatives, there is a growing exploration for raw materials capable of replicating meat products. Chia seeds emerge as a promising candidate. Defatted chia flour, known for its high protein content, is valuable, especially due to the chia mucilage fraction. This can serve as an effective ingredient contributing to enhancing the textural properties of plant-based meat alternatives [324]. Quinoa and chia, as viable options for reintegrating into the food value chain, offer coproducts that, when utilized in meat production innovation, especially in controlling lipid oxidation, can play a crucial role in enhancing shelf life [325,326].

Amaranth, with its distinctive earthy and peppery flavour profile, has become a culinary cornerstone in Central and South America, India, and Africa. In Mexico, it is favoured for crafting delectable, sweet treats, while in India, it features a myriad of both savoury and sweet dishes. Its minuscule grains contribute a unique texture and nutritional richness to a diverse range of culinary creations, including soups, stews, and baked goods. Recently, there has been a resurgence in the popularity of amaranth, marking a revival five centuries after it served as a staple food for ancient Mesoamericans [327]. Today, amaranth is easily accessible in a variety of commercial food products worldwide, showcasing its enduring popularity and recognized nutritional benefits [328]. Recent developments involve integrating grain amaranth into energy foods, including breakfast items, bread, crackers, and pancake mixes. Additionally, it can be enjoyed popped as a wholesome snack option [327].

The scientific literature robustly supports the culinary utilization of *Moringa oleifera*, highlighting its potential as a valuable source of phytochemicals with diverse applications in functional and medicinal foods [329,330,331,332]. The incorporation of moringa into bakery products has shown significant nutritional benefits, such as improved digestibility, enhanced dough stability, increased antioxidant capacity, and extended preservation [333].

Additionally, moringa’s antioxidant and antimicrobial properties make it suitable for reducing lipid oxidation and inhibiting bacterial growth, thereby enhancing the nutritional content of food items like fish or chicken burgers, cream cheese, or mutton patties [334,335,336]. Protein extracts derived from fractions of moringa seeds have been reported as a potential ingredient to uphold texture and reduce syneresis in yoghurt production [337]. As the trend in food fortification is increasing, moringa has been explored as a functional ingredient in fortification in protein and fibre with acceptable sensory quality in products such as pasta, cookies, bread, or tempeh-like fermented products [338,339,340]. Incorporating moringa into the food industry offers a significant opportunity to tackle nutritional deficiencies, particularly in regions with extreme poverty or food scarcity [341]. By harnessing moringa’s nutritional richness, food manufacturers can develop products that provide nourishment and enhance the well-being of communities in need [342].

Also, buckwheat, millet, and teff are noteworthy emerging crops, particularly as gluten-free grains, making them suitable for food industrial applications that prioritize nutritious and gluten-free food products [343]. Buckwheat is experiencing increased popularity in Eastern Europe, featuring in dishes like blini, and Asian cuisine through soba noodles. Its nutritional value and presence of bioactive compounds like peptides, flavonoids, and phenolics position it as a promising crop for developing gluten-free products such as cookies, bread, or focaccia to cater to individuals with celiac disease [344,345]. In Central Asia, where buckwheat is a dietary staple, a strategy to decrease meat consumption involves introducing 3D food printing in the formulation of gluten-free, plant-based meat alternatives creating and customizing balanced foods [346]. Aligned with buckwheat, millet, a dietary staple in many African and Asian nations, is utilized in porridges, flatbreads, and pilafs, and its gluten-free nature also makes it a suitable alternative for individuals with gluten sensitivities [345,347]. Also, the increasing global demand for fermented foods has opened the possibility of utilizing multigrain millet, given its substantial probiotic properties [348]. For instance, fermenting whey cereal beverages with millet and moth bean produced an innovative dairy alternative with desirable functional and sensory traits, maintaining acceptability for 12 days [349]. Originating from Ethiopia, teff flour enhances the technological process and quality of wheat bread, boosting its nutritional value and specific qualities by improving structural and mechanical parameters [350]. Additionally, teff serves as a substitute for breadcrumbs and soybean meals in conventional meat products, enhancing moisture retention and sensory characteristics, making it suitable for individuals with celiac disease and soy protein sensitivity [351]. Furthermore, extrusion cooking has been investigated to create expanded teff-based products mixed with chickpea flour, showing promising physical, functional, and sensory attributes, indicating potential applications in the food industry [352].

Hemp seeds, known for their nutty flavour, play a prominent role in global cuisine, whether sprinkled over salads, blended into smoothies, or incorporated into baked goods contributing to a boost in protein and omega-3 fatty acids. Various processes for producing value-added ingredients from hemp seeds, including oil and protein, are well suited for targeted food applications, presenting market-led opportunities. Examples include hemp flour and derived oil for fortified food products, as well as the extraction of protein isolates and concentrates for plant-based meat and dairy alternatives [353]. For instance, incorporating hemp flour in cookies, which imparts high antioxidant activity, or substituting starch with 20% hemp protein concentrate to enhance the quality of gluten-free bread [354,355] or fermenting hemp seeds to produce fermented drinks with prebiotic activity, has resulted in an alternative dairy product [356]. Furthermore, both dry and high-moisture extrusion techniques have resulted in hemp-based products, including energy bars and meat analogue formulations, exhibiting high sensory acceptability [313,357].

Finally, in Middle Eastern cuisine, desert truffles are revered as a delicacy, imparting a distinctive earthy and umami flavour to dishes. They are savoured through various preparations, including frying, boiling, or as a meat replacement in cooked dishes [155]. From rice-based delicacies to sauces and desserts, the adaptability of desert truffles showcases their enduring appeal in both traditional and contemporary culinary settings [315]. Due to the nutritional profile of this crop, which encompasses a wide range of bioactive chemical constituents, it has been utilized as a therapeutic source [358].

#### 3.5.2. Technological Treatments

Despite their increasing use in the food industry, minor cereals and pseudocereals often have poor functional properties for certain food manufacturing processes. To overcome these shortcomings, flours or grains of these species have been subjected to various technological treatments that induce structural modifications of their main biopolymers (starch and proteins), leading to significant improvements in their techno-functional properties. Among the technological treatments applied, the physical modification of starches, flours, and grains has become increasingly important, as it allows the functional improvement of these matrices while maintaining the status of food ingredients so that they can be incorporated into the formulation of clean label foods.

Heat Moisture Treatment (HMT) is one of the most commonly used and is carried out at temperatures above the starch gelatinization temperature while maintaining a moisture content of less than 35%. HMT has been applied to buckwheat starch and flour to improve their thermal stability and nutritional properties, making them more suitable for pasta and baked products [359,360]. The treatment of quinoa with HMT has also been proposed to modify the gelatinization behaviour, relative crystallinity, and Fourier Transform Infrared (FTIR) spectra of starch, improving its range of applications as a food thickener [361], and to increase starch/protein interaction in flours, thereby reducing the estimated glycaemic index [362]. Microwave-assisted hydrothermal treatment (MWT) has emerged as a promising alternative, as it allows for faster treatment and reduces the amount of energy required in comparison to conventional thermal heating. Studies on buckwheat grains have shown that the moisture content during MWT modulates the final properties of the flour, thus widening the range of applications for buckwheat in the food industry, such as soups and sauces, cakes, frozen desserts, and bakery products [363]. The use of MWT-treated buckwheat flour in the production of gluten-free products resulted in improved dough viscoelasticity and bread-specific volume, delayed staling, and modulated glucose release kinetics [364]. In the same way, MWT is effective in modifying the functional properties of quinoa flours by altering their protein and starch components, thus influencing the rheological behaviour of bread doughs, which affects breadmaking performance [365].

Annealing (ANN) is an alternative hydrothermal treatment in which starchy materials are treated in excess water between the glass transition and gelatinization temperatures. This technology has been successfully used to improve the thermal stability of buckwheat starch while modifying its in vitro digestibility, increasing slow digestible starch and resistant starch [366]. Previous studies on the ANN treatment of amaranth starch have shown that the formation of cross-links between starch chains (amylose–amylopectin) alters the physicochemical properties of amaranth starch with improved thermal stability, increasing its potential use as an ingredient in the food industry [367].

High-intensity ultrasound (US) has also been used for the treatment of starches and flours, which are generally suspended in an aqueous medium in which they undergo physical modification by the phenomenon of cavitation. This technology has been applied to minor cereals such as canary seeds to improve the functional properties of the flours [368]. Sonicated defatted canary seed flours proved to be versatile ingredients with desirable surfactant properties for a wide range of applications such as cakes, muffins, and bread where texture enhancement, stability, and consistency are critical.

High hydrostatic pressure (HHP) is a technology generally used in the treatment and preservation of foodstuffs, which has recently been applied to the treatment of cereals and pseudocereals due to its ability to exert significant effects on starch and protein polymers. The ability to modify the hydration and pasting properties of quinoa and buckwheat starches, as well as the texture and rheology of the gels formed from them, has recently been investigated [369,370,371]. HHP treatments have also been applied to teff and buckwheat flours [372] and whole buckwheat grains [373], improving the phenolic content of the resulting flours and their structuring capacity in gluten-free systems, depending on the treatment variables.

## 4. The Health Benefits of NUSs for the Mediterranean Region

Globally, developed countries are experiencing an increase in the incidence of major non-communicable diseases such as cardiovascular disease, diabetes, and obesity. This increase has led to a significant rise in morbi-mortality and healthcare expenditure [374]. The search for novel foods that not only provide balanced nutrition but also have multiple beneficial health effects has made the study of functional foods an incipient field. In this context, the health-promoting aspects of bioactive compounds found in NUS, such as quinoa, chia, amaranth, hemp, moringa, sorghum, buckwheat, teff, or dessert truffles, have been elucidated. Quinoa and amaranth, in particular, not only have commendable nutritional profiles but are also significantly richer in both micro- and phytonutrients than conventional grains, which has led to a growing interest in these crops for potential integration into functional foods and nutraceuticals [375]. Recent studies have focused on quinoa and amaranth intending to identify numerous bioactive compounds, including phenolic acids, flavonoids, betalains, carotenoids, and tocopherols. These compounds have been found to have significant antioxidant, -inflammatory, -hypertensive, and -hyperglycaemic and lipid-lowering properties, and are the main mechanistic basis for their beneficial effects against cancer, neurodegeneration, and metabolic disorders such as obesity and diabetes, which are major risk factors for cardiovascular disease [375]. By and large, quinoa and amaranth are generally considered safe for consumption and are not commonly associated with allergies, although isolated cases have been reported [376]. However, more research is needed to fully understand the prevalence and mechanisms of amaranth and quinoa allergy.

Chia seeds also have excellent nutritional properties [150]. In addition to their high fibre content and balanced range of macro- and micro-nutrients, chia seeds are known for their high concentration of α-linolenic acid, which serves as a precursor for the synthesis of ω-3 fatty acids with long chains such as DHA and EPA. It is now well established that omega-3 polyunsaturated fatty acids (PUFAs) confer several health benefits. In particular, the consumption of omega-3 fatty acids has been associated with a decrease in triglyceride and non-high-density lipoprotein cholesterol levels, supporting the protective role of these fatty acids against cardiovascular events [377]. In addition to reducing dyslipidaemia, omega-3 fatty acids also exert protective benefits against atherosclerosis, inflammation, diabetes, and cancer, although some research findings are controversial and highlight the need for more clinical studies to clarify the beneficial effects of ω-3 PUFAs [378]. Chia seeds provide health benefits due to bioactive compounds such as phenolic molecules, tocopherols and carotenoids. These compounds have antioxidant properties that contribute to health benefits, particularly in the fight against diseases such as diabetes and cancer [224]. Chia seeds also contain phytosterols, which are mainly associated with reduced cholesterol and hepatoprotective effects. They also play a beneficial role in diabetes and cardiovascular diseases [241].

In recent years, the distinctive components and biological activities of moringa have been extensively studied and documented, highlighting its potential health benefits and therapeutic applications [379]. The whole *M*. *oleifera* plant, including leaves, fruits, pods, flowers, seeds, and roots, exhibits a range of biological activities. Leaves are rich in active constituents such as flavonoids, polyphenols, terpenoids, phenylpropanoids, fatty acids, sterols, and alkanes, as well as vitamins and minerals [380,381,382,383]. Among these, flavonoids and polyphenols are the primary bioactive molecules, showing antioxidant, -cancer, -sepsis, -inflammatory, -hypertensive, and -diabetic and hepatic lipid-lowering properties and beneficial effects on obesity-related reproductive diseases. In addition to fatty acids, essential amino acids, and flavonoids, the seeds also contain isothiocyanates, which play an important role as anti-inflammatory, -oxidant, -bacterial, and -cancer agents [384]. Furthermore, newly discovered components extracted from the stem of the plant, such as tricosanoic acid, cholest-5-en-3-ol, stigmasterol, and gamma-sitosterol, have shown potent antifungal activity [379]. Potent antioxidants such as tocopherols, ascorbic acid, flavonoids, and carotenoids have also been extracted from *M*. *oleifera* flowers [385].

Hemp seed, derived from the plant *Cannabis sativa* L., has a rich history of use in Asia dating back to prehistoric times. Recently, countries such as the United States, Canada, and Australia have now legalized the farming and use of hemp seed, provided that its tetrahydrocannabinol content remains below 0.3%. This legalization has increased interest in hempseed due to its recognized nutritional benefits and perceived pharmaceutical applications [386,387]. Historically, the economic importance of hemp has been in the use of the fibre-rich stem, valued to produce textiles, clothing, and paper goods, while its seeds have remained largely underutilized. However, in recent years there has been a surge of interest in exploring the nutritional and pharmaceutical properties of hemp. In particular, hempseed oil has the highest concentration of PUFAs of any vegetable oil. It is widely accepted that an increased intake of PUFAs is associated with a reduced risk of several health conditions, including cardiovascular disease, cancer, rheumatoid arthritis, hypertension, inflammatory disorders, and autoimmune diseases. Of note is the highly desirable *ω*-6:*ω*-3 ratio found in hempseed, which typically ranges from 2.5 to 3.5/1, in contrast to the *ω*-6:*ω*-3 ratio found in typical Western diets, which normally ranges from 15 to 17/1. The importance of the *ω*-6/*ω*-3 ratio in human health is well documented, with a high ratio (>10:1) in the human diet being associated with an increased risk of cancer, inflammation, and cardiovascular disease. On the contrary, a ratio lower than 2.5:1 has been associated with a reduced risk of chronic diseases, the suppression of cancer cells, and lower mortality [184]. In addition to its oil content, cannabis seeds contain two different phenolic compounds: lignanamides and hydroxycinnamic acids. In vitro studies have extensively documented the antioxidant properties of hempseed lignanamides, including compounds such as N-trans-caffeoyltyramine and cannabisin B. Additionally, hempseed is rich in flavonoids, which contribute significantly to its antioxidant capabilities, as demonstrated by cell-free system assays such as ORAC and FRAP. Furthermore, the different fractions obtained from hempseed processing show different levels of antioxidant activity, with the coarsest fraction, dominated by hulls, showing the highest potency in this regard [388]. Hemp seed also contains a higher concentration of tocopherols than most vegetables. These tocopherols have antioxidant properties that effectively inhibit the oxidation of unsaturated fatty acids [389]. Furthermore, the potential health benefits of cannabinoids, including CBD, have been explored in relation to various conditions such as Alzheimer’s disease, cancer, epilepsy, inflammatory diseases, Parkinson’s disease, and amyotrophic lateral sclerosis, as observed in studies conducted in mice [390]. In general, preclinical studies indicate the beneficial effects of hempseed supplementation on improving the blood lipid profiles and levels of linoleic acid, α-linolenic acid, and γ-linolenic acid. These enhancements have been associated with improvements in metabolic syndrome and neurodegenerative diseases [391]. In human trials, while several studies have described the beneficial effects of hempseed oil on skin health, mental well-being, and neurological disorders, the effects of hempseed on cardiovascular disease require further verification. Existing studies have produced conflicting results, indicating the need for further research (revised by [386]).

The health benefits of foods made from sorghum grain lie in the fact that sorghum is a rich source of antioxidant compounds such as vitamins, as well as macro- and micro-nutrients, including phenolic acids, flavonoids, and sterols [392]. The high antioxidant activity of bioactive compounds in sorghum grain is mainly due to its polyphenolic composition. In particular, white sorghum flour contains almost twice as many polyphenols as red sorghum flour. Numerous studies have demonstrated the anti-obesogenic effects of sorghum. Extracts of this grain have been shown to significantly inhibit the differentiation and accumulation of triglycerides. In addition, sorghum-supplied diets have been shown to reduce body weight in rats [393]. Human clinical trials have been conducted with sorghum to investigate obesity-related parameters. Participants who consumed sorghum biscuits reported greater satiety than those who consumed wheat-based products [394]. Additionally, sorghum starch undergoes slower digestion compared to starch from other cereals because of the hard outer layer of the endosperm and the existence of tannins. As a result, findings from these investigations indicate the possible incorporation of sorghum into diets for weight control.

Buckwheat is rich in several bioactive compounds that, along with essential nutrients, contribute to positive health outcomes [51]. More than a hundred bioactive compounds have been identified in buckwheat [144], including high levels of flavonoids such as rutin (quercetin-3-d-rutinoside), epicatechin, and quercetin, which vary depending on the type and origin of the buckwheat [395,396,397]. Moreover, the level of phenolic compounds is highly dependent on the roasting process, which can affect the total content and lead to significant losses [398]. Although the available literature indicates that buckwheat supplementation may provide some benefit in lowering TC and glucose in the context of dyslipidaemia and type 2 diabetes, a recent systematic review and meta-analysis have shown a small association between buckwheat supplementation intervention and cardiovascular risk factors [399]. Therefore, future human trials are needed to unveil the beneficial effects of the bioactive compounds in buckwheat.

Teff, another gluten-free pseudo-cereal like quinoa and buckwheat, has attracted considerable research interest due to its many potential health benefits. The use of teff is an opportunity to address the nutritional shortcomings of current commercial gluten-free products. Rich in insoluble polysaccharides, as well as macro- and micro-nutrients, teff is being explored as a viable option for people with type 2 diabetes due to its low glycaemic index. Teff is also naturally high in iron, making it suitable for people with anaemia [400]. Teff also contains higher levels of minerals such as zinc, iron, and calcium than other grains [401]. Furthermore, ferulic acid, a phenolic acid compound known for its potent antioxidant properties, is a major component of teff, along with other bioactives such as flavones like apigenin-6,8-c-diglucoside and apigenin-8-c-glucosyl-7-O-glucosides [400].

Furthermore, many of these NUSs contain antinutritional compounds such as phytic acid and saponins, which have the potential to bind nutrients and reduce absorption in our bodies. However, the use of various processing techniques such as fermentation, sprouting, extrusion, and cooking can increase their bioavailability [375].

Finally, it is important to emphasize the importance of bioactive peptides derived from the NUSs. These peptides, typically 2 to 50 amino acids in length, remain inert within the intact protein. However, when released by proteases during digestion or through external hydrolysis, they are absorbed by the intestinal tract, reach various organs, and modulate important biological activities. These activities include immunomodulatory, antioxidant, antihypertensive, hypolipidaemic, hypoglycaemic, and hypotensive effects. Such effects play a key role in metabolic diseases such as obesity, metabolic syndrome, or metabolic-associated fatty liver disease [402,403,404,405]

## 5. Future Prospects

In the context of global economic and climatic change, a deeper understanding of agricultural production intensification is required, and new opportunities and challenges must be identified to progress [406]. Following high-input resource-intensive farming systems, an innovative approach is needed to protect and enhance natural resources while increasing productivity. Emerging crops appear capable of ensuring future food security and sovereignty [407]. They contribute to climate change mitigation and adaptation, demonstrating resilience and adaptation to abiotic stress [54,408,409,410] For instance, salinization and drought are two major factors altering conditions for crop growth. Quinoa, a facultative halophyte, can withstand high salinity levels and water scarcity [411], and amaranth enriches its nutritional content and antioxidant activity under stress conditions [412]. Furthermore, chia has proven to be a viable option in dry, sandy soil and high-temperature environments [412], growing successfully in water-limiting situations [413], as does desert truffle [41]. Additionally, these crops require lower inputs for productivity, resulting in the limited use of water, fertilizers, and pesticides [33,414,415]. This makes NUSs pivotal in reducing greenhouse gas emissions [416] and promoting the sustainable development and efficient management of natural resources such as water, soil, and air. Moreover, NUSs promote soil protection, aligning with the objectives of the EU policy (CAP 2023–27), aiming for a 20% reduction in fertilizer use by 2030 and the potential to preserve and improve soil health [412].

NUSs also hold the potential for phytoremediation, offering a feasible option to mitigate the effects of heavy metal stress. Different species exhibit varying degrees of heavy metal accumulation and tolerance mechanisms. Using these species as a phytoremediation approach is seen as an optimal solution for reclaiming soil contaminated with heavy metals, ensuring the safety of harvested crops for human consumption. Quinoa, for instance, has shown promising bioremediation potential for soil contaminated with heavy metals [417], while amaranth has been studied extensively for remediating cadmium (Cd)-contaminated agricultural soils [418].

Therefore, meeting the needs of a growing world population by 2050 will require more resources from intensive food production. Monocultures may be vulnerable to unpredictable climate change, potentially leading to severe productivity losses. As previously mentioned, crop diversification, especially through the rotation of underutilized crops with existing systems, can improve food security, disrupt disease and pest cycles, replenish soil nutrients, and diversify pollinator presence [407]. Additionally, the increasing demand for ingredients used in animal feed creates pressure on major crops. In line with this, NUSs can serve as alternative sources of energy and protein, reducing dependence on major crops and competition with human food purposes. These crops could play a crucial role in animal growth and productivity due to their nutritional qualities. For example, Amaranth leaves and grains have shown varied nutritional qualities depending on consumption by monogastric or ruminants [419,420,421]. Quinoa, with its protein-rich fodder, offers an opportunity to diversify its use beyond animal feed.

Thus, NUSs have the potential to play a crucial role in addressing the world’s current food challenges. They offer sustainability, adaptability, and nutritional value that can contribute to food security, conserve agrobiodiversity, and make agricultural and agrifood value chains more resilient. Despite limited progress in the development of superior underutilized crops, greater efforts will be needed to integrate these crops into mainstream agriculture.

## 6. Conclusions

In conclusion, the exploration of emergent crops in the dynamic context of agriculture and food science presents a groundbreaking avenue towards sustainable cultivation and nutritional innovation, with profound implications for human health. This review has delved into the potential of utilizing NUSs in Mediterranean environments, shedding light on the manifold benefits of diversifying agricultural and food systems. The resilient nature, nutritional richness, and positive health impacts of NUS such as quinoa, amaranth, chia, moringa, buckwheat, millet, teff, hemp, and desert truffles have been thoroughly examined, emphasizing their adaptability to the changing climate. By embracing these crops, we unlocked novel opportunities for agriculture and functional food development in important susceptible agricultural areas such as the Mediterranean region, paving the way for a more resilient and adaptable food system. The analysis made extends beyond the agronomic and nutritional aspects, delving into the broader dimensions of environmental, biomedical, economic, and cultural significance. Promoting agricultural diversification emerges as a key strategy to enhance food system adaptability to evolving environmental conditions, fostering sustainability and resilience. Therefore, by integrating these crops into our agrifood systems, we not only bolster agriculture resilience but also enhance food quality, addressing a spectrum of dimensions crucial for sustainable development. In essence, this revision calls for a paradigm shift towards embracing emergent crops, highlighting their potential to redefine our approach to agriculture, food science, and health. The integration of neglected and underutilized species into our food systems represents a forward-thinking solution, steering us towards a more sustainable and resilient future.

## Figures and Tables

**Figure 1 plants-13-01914-f001:**
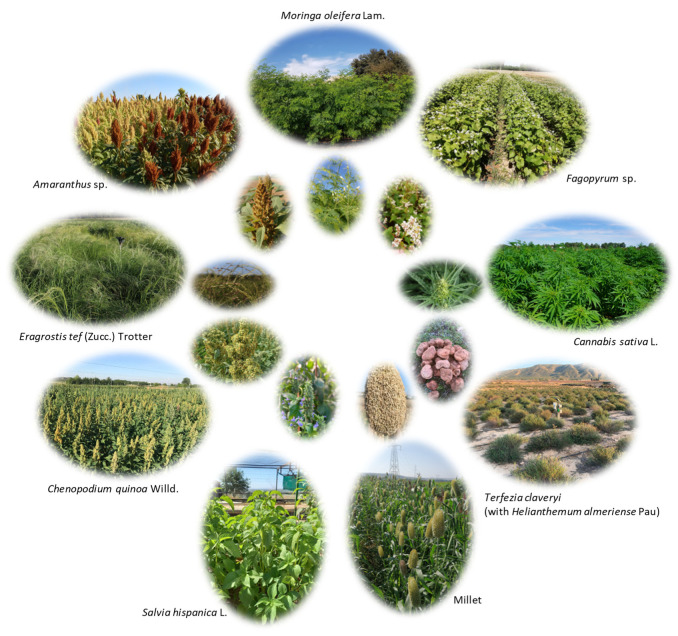
NUS crops with the potential for cultivation in Mediterranean environments. Representative images of different nutrient-rich and stress-resilient emergent crops for sustainable and healthy food in the Mediterranean region, grown under field conditions across different regions of Spain. The images depict the crops in the field (outer images) and the panicle, inflorescence, or truffle for each crop (inner image). The crops included in the figure are *Chenopodium quinoa* Willd., *Amaranthus* sp., *Fagopyrum* sp., *Eragrostis tef* (Zucc.) Trotter, *Salvia hispanica* L., *Moringa oleifera* Lam., *Cannabis sativa* L., and millet (*Pennisetum glaucum* (L.) R.Br. *Terfezia claveryi* Chatin (with *Helianthemum almeriense* Pau)).

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
