# Peer review of "From ‘Farm to Fork’: Exploring the Potential of Nutrient-Rich and Stress-Resilient Emergent Crops for Sustainable and Healthy Food in the Mediterranean Region in the Face of Climate Change Challenges"

_plants, 2024, doi:10.3390/plants13141914_

Round 1

Reviewer 1 Report

Comments and Suggestions for Authors

The work entitled "From `Farm to Fork´: Exploring the Potential of Nutrient-Rich and Stress-Resilient Emergent Crops for Sustainable and Healthy Food in the Mediterranean Region in the Face of Climate Change Challenges" is a very comprehensive collection of information on NUS crops. Without a doubt, it is a very interesting work, and the Authors have shown great insight into the described issue. My concern, however, is the comprehensiveness of this work. Not counting the bibliography list, the work is nearly 43 pages long. Which, even for review papers, is quite unusual. As a result, the potential reader may not be interested in familiarizing himself with the entire text. My suggestion to the Authors is that this work be divided into two parts. The first part could focus on the potential of NUS plants in agroecosytem, climatic requirements, that is, all the information that is included up to page 16. And the second manuscript could deal with nutritional aspects. This is only a suggestion on my part for the Authors' consideration.
However, if the paper was not split. I propose to reduce the volume of the manuscript by shortening, first of all, the part devoted to the description of the chemical composition of NUS plants. Then the Authors' attention should be focused on the most important nutrients such as proteins, fats, carbohydrates, anti-nutrients and antioxidant compounds.
In addition, there are minor errors in the text, which I point out below:
1. Line 60 - it is worth mentioning what crops are mainly concerned.
2. the scientific name of the species consists of two parts - the name of the genus and the name of the species (species epithet), accompanied at the end by a third element - the name of the author(s) of the name. In the work, the last third element is missing at Latin names. For example, line 159 is Pisum sativum, and should be Pisum sativum L.
3. Line 441 - citation should be corrected. Comply with the requirements of the journal.
4. line 513 - "in A. hypocondriacus[141]" no spaces.
5. Line 518-520 - italics missing from Latin names.
6. Line 674-675 - citation should be corrected.
7. Throughout the manuscript, citations are listed after a comma in separate square brackets (example [4], [5], [6].). A more correct notation would be this form [4,5,6].

Author Response

Dear Reviewer 1,

first, we would like to express our sincere gratitude for your dedicated time and effort in reviewing our work. Your insights are highly appreciated, and we have tried to incorporate all your suggestions into this revised version of the manuscript.

Regarding your comment related to the extension, we did contemplate dividing this work into two parts (Part I and Part II) due to its extensive nature. However, given that we were covering seven different crops, some of which encompass various plant species and even genera, and aiming to provide a comprehensive overview from agronomical, food science and technology, and biomedical perspectives, we decided to consolidate all the information into a single manuscript. While this approach resulted in a manuscript with a large number of references, we believed it was preferable to ensure both parts were published in parallel. This decision was motivated by our desire to maintain coherence across the various perspectives presented.

Nonetheless, and taking into account your feedback and that of Reviewer 2, which also addressed this point, we have revised the text accordingly. Specifically, in response to your suggestion, we have eliminated certain sections that were more generalized, such as discussions on some bioactive compounds. Instead, we have concentrated the manuscript on the most pertinent information about the NUSs studied here.

In addition, there are minor errors in the text, which I point out below:
1. Line 60 - it is worth mentioning what crops are mainly concerned.

Thank you, Reviewer 1. We have incorporated the three most cultivated crops into Lines 60-61.

  1. the scientific name of the species consists of two parts - the name of the genus and the name of the species (species epithet), accompanied at the end by a third element - the name of the author(s) of the name. In the work, the last third element is missing at Latin names. For example, line 159 is Pisum sativum, and should be Pisum sativumL.

Thank you, Reviewer 1, we are sorry about that, you are right. We have revised the text accordingly and changed it to the trinomial nomenclature as suggested.

  1. Line 441 - citation should be corrected. Comply with the requirements of the journal.

Thank you, Reviewer 1. The format has been corrected.

  1. line 513 - "in A. hypocondriacus[141]" no spaces.

Thank you, Reviewer 1. A space has been included.

  1. Line 518-520 - italics missing from Latin names.

Thank you, Reviewer 1. This has been corrected accordingly.

  1. Line 674-675 - citation should be corrected.

Thank you, Reviewer 1. This has been changed.

  1. Throughout the manuscript, citations are listed after a comma in separate square brackets (example [4], [5], [6].). A more correct notation would be this form [4,5,6].

We apologize, this has been a problem with the Reference editor used. We have corrected this aspect as requested in the final version (document without track changes) uploaded into the system.

Reviewer 2 Report

Comments and Suggestions for Authors

The manuscript “From `Farm to Fork´: Exploring the Potential of Nutrient-Rich and Stress-Resilient Emergent Crops for Sustainable and Healthy Food in the Mediterranean Region in the Face of Climate Change Challenges” is an interesting idea. The manuscript is written well, however, it needs few changes before any decision.

The manuscript is too long which can divert the attention of readers. Therefore, authors must take care of this issue. Please carefully revise the manuscript and just focus on to give clear message to reader rather than a lengthy and laborious work.

What is the importance of emergent crops? This information can be added in the abstract section.

What outbreak this article will bring? Authors can add this information in revised abstract section.

The section 1.1 “Historical Background” can be deleted from the text, as this section is not mandatory.  

There is no figure of all the mentioned crops in the text. Therefore, I suggest the authors to add figures of all the crops it will give a good message to readers.

At the end of manuscript, authors only added the conclusion section. Here authors must add the future prospective and challenges that need to be addressed for ensuring the better production of these crops.

Comments on the Quality of English Language

Need to go through for typos and grammatical errors.

Author Response

Dear Reviewer 2,

we would like first to extend our sincere gratitude for your thorough review and valuable feedback on our manuscript. Your insights have significantly contributed to the improvement of our work. We truly appreciate your time and effort to evaluate our revision.

Thank you as well for your thoughtful comment regarding the length of our manuscript. We carefully considered the option of splitting the work into two parts (Part I and Part II) due to its extensive coverage. However, given the comprehensive nature of our study, which encompasses seven different crops, including various plant species and even genera, and aims to provide a holistic overview from agronomical, food science and technology, and biomedical perspectives, we ultimately chose to consolidate all the information into a single manuscript. While this decision resulted in a manuscript with a large extension and a larger number of references, we believed it was important to maintain coherence across the different perspectives presented. Ensuring that both parts are published in parallel allows for a more cohesive understanding of the topic. We hope this clarifies our rationale behind the manuscript structure. Now, considering your comment, we have gone through the text deleting those parts that were not that relevant to the topic (when were too general) in order to keep the main messages clear (the information that involved the characteristics and properties of the seven NUS crops with the potential to be implemented in the Mediterranean Agrosystems.

Regarding the two points made for the abstract, “What is the importance of emergent crops?” we have included this information “Emergent crops offer opportunities for diversifying agriculture, bolstering food security, and creating economic prospects amid evolving environmental and market conditions” (Lines 40-42) and regarding “What outbreak this article will bring?” we have summarized it in the Abstract including this information “Agricultural diversification in the Mediterranean region can enhanced sustainability, resilience, and food security, thereby mitigating the risks associated with monoculture practices and bolstering local economies and livelihoods under new climate scenarios”.

Regarding the comment concerning section 1.1 "Historical Background," we have opted to keep this section as we consider it integral to conveying the central message of our work. Historically, the introduction of new crops has played a crucial role in fortifying the agrifood system within the Mediterranean region. We believe that maintaining this section aligns with our approach to enriching our agriculture and food systems through historical context and perspective. Nonetheless, we have revised this section and deleted the parts that were less relevant to the topic.

Thank you, Reviewer 1. Your suggestion about including a Figure with pictures of the crops here studied will increase the quality of the manuscript. We have included this Figure as Figure 1, with pictures of the studied crops in the field frown under Mediterranean conditions.

Last but not least, concerning the future prospects, we have included this section right before Conclusions, just to make clear to the reader these important aspects, as Reviewer 2 recommended.

Need to go through for typos and grammatical errors.

Thank you, Reviewer 2, we are sorry about that, you are right. There were many typos and errors (including in the format of some references) and we have revised the text and corrected it accordingly.

Round 2

Reviewer 1 Report

Comments and Suggestions for Authors

The manuscript has been revised according to all comments. The authors have also reduced the volume of the text. Taking the above into account, I believe that the manuscript in its present form can be published.

Author Response

Dear Academic Editor,

Thank you for your kind words and for acknowledging our efforts in preparing this Review. We greatly appreciate your positive feedback and your valuable suggestions for improvement.

We understand the necessity of adhering to the journal's guidelines regarding manuscript length and the number of references. The comprehensive nature of our review, which covers agronomic, nutritional, technological, and medicinal aspects of seven different emergent crops, posed a unique challenge in balancing depth and conciseness.

In response to your comments, we have carefully revised the manuscript, striving to be more concise. We have reduced the length by condensing paragraphs, removing non-essential sentences, and limiting the number of examples provided. However, we believe that further reductions would compromise the focus and relevance of our review. Given the complexity and interdisciplinary nature of the topic, each section is essential to provide a comprehensive understanding to the readers of Plants-MDPI.

While we have made significant reductions, we have ensured that the manuscript retains its informative and integrative nature. We hope that the current version meets the journal's expectations while maintaining the depth and breadth necessary for such a multifaceted topic.

Attached, you will find the revised version with track changes activated, so you can easily see the changes and improvements made. A clean version of the manuscript will be uploaded upon acceptance.

Thank you once again for your constructive feedback and encouragement. We look forward to your understanding and consideration of the revised manuscript.

Sincerely,

Maria Reguera
